# HeteroTCR: A heterogeneous graph neural network-based method for predicting peptide-TCR interaction
Zilan Yu [1,2,5], Mengnan Jiang[1,5] & Xun Lan [1,2,3,4] ✉

Identifying interactions between T-cell receptors (TCRs) and immunogenic peptides holds profound implications across diverse research domains and clinical scenarios. Unsupervised clustering models (UCMs) cannot predict peptide-TCR binding directly, while supervised predictive models (SPMs) often face challenges in identifying antigens previously unencountered by the immune system or possessing limited TCR binding repertoires. Therefore, we propose **HeteroTCR**, an SPM based on Heterogeneous Graph Neural Network (GNN), to accurately predict peptide-TCR binding probabilities. HeteroTCR captures within-type (TCR-TCR or peptide-peptide) similarity information and between-type (peptide-TCR) interaction insights for predictions on unseen peptides and TCRs, surpassing limitations of existing SPMs. Our evaluation shows HeteroTCR outperforms state-of-the-art models on independent datasets. Ablation studies and visual interpretation underscore the Heterogeneous GNN module's critical role in enhancing HeteroTCR's performance by capturing pivotal binding process features. We further demonstrate the robustness and reliability of HeteroTCR through validation using single-cell datasets, aligning with the expectation that pMHC-TCR complexes with higher predicted binding probabilities correspond to increased binding fractions.

T cells are a vital component of the adaptive immune system. T-cell receptors (TCRs) present on the surface of T cells have the ability to recognize peptides presented by Major Histocompatibility Complex (MHC) derived from various sources such as host proteins, pathogens, or tumors[1]. A majority of TCRs consist of a pair of $\alpha$- and $\beta$-chains. During the maturation of T cells, TCR undergoes random rearrangement of *variable* (V), *diversity* (D), and *joining* (J) gene segments[2]. The interaction between TCR and peptide-MHC (pMHC) complex is mainly defined by three complementarity-determining regions (CDRs) for each chain[3]. Since the $\beta$-chain contains V-, D-, and J genes with higher diversity and the CDR3 regions are the determinant of the specific recognition of peptide[4], most studies have focused on CDR3 regions of TCR $\beta$-chain[5].

Reliably predicting peptide-TCR binding solely from sequences is of great significance for cancer immunotherapy and vaccine development[6]. Such predictions also provide valuable insights into patients' immunological history, including responses to immunization, infections, and vaccines, thus paving the way for personalized healthcare[7]. Biological experiments to screen numerous unfiltered candidate immunogenic peptides are expensive and time-consuming[8]. Therefore, developing an effective computational

method to predict peptide-TCR interactions is pressing due to the intricate nature of the molecular-level process. Notably, a peptide can bind to different TCRs, and conversely, a TCR is able to recognize multiple peptides[9].

Existing models for predicting peptide-TCR binding can be divided into two categories[10]: (1) Unsupervised clustering models (UCMs) group similar TCR sequences together into clusters based on their shared features, including GLIPH[11], TCRdist[12], GIANA[13], and iSMART[14]; (2) Supervised predictive models (SPMs) can be further classified into categorical epitope models and generic models. Categorical epitope models are designed to learn TCR patterns that are specific to a particular peptide by using antigen-specific labels[15]. Examples of such models are TCR-classifier[16], TCRex[17], TCRGP[18], and DeepTCR[19]. In contrast, generic models, such as NetTCR-1.0[20], NetTCR-2.0[5], ERGO Long Short Term Memory (LSTM) and Auto-encoder (AE)[21], DLpTCR[22], ImRex[23], and TITAN[7], can predict interactions between any TCR and peptide, without being restricted to a specific peptide.

Though UCMs have shown promise in various applications, they are not directly applicable to recognizing peptide-TCR binding[24]. Additionally, categorical epitope models require training for each peptide or set of peptides in multiclass classification, demanding ample TCR training data

¹School of Medicine, Tsinghua University, 100084 Beijing, China. ²Centre for Life Sciences, Tsinghua University, 100084 Beijing, China. ³Tsinghua-Peking Center for Life Sciences, MOE Key Laboratory of Tsinghua University, Beijing, China. ⁴MOE Key Laboratory of Bioinformatics, Tsinghua University, 100084 Beijing, China. ⁵These authors contributed equally: Zilan Yu, Mengnan Jiang. ✉e-mail: xlan@tsinghua.edu.cn

binding to the same peptide[23]. Furthermore, these models are inherently incapable of predicting the interactions of peptides not included in the training set (unseen epitopes)[7]. Existing generic models predict interactions between any TCR and peptide, without being restricted to a specific peptide, but rely on concatenating two interacting sequences encoded with a BLO-SUM matrix or physicochemical properties[23]. These approaches might not directly focus on the interaction problem, instead learning embeddings for individual interactors, resulting in diminished model performance[23]. Moreover, although these studies perform well on the testing set that includes the training set peptides or TCRs, they cannot reliably predict interactions for those not present in the training set[6]. Additionally, they lack generalizability to new data from diverse databases[22], due to the extensive variation in TCR and peptide sequences[23].

Graph neural networks (GNNs) have made remarkable strides in recent years, establishing themselves as essential tools for a range of graph-based applications. Notably, they have demonstrated success in predicting chemical stability[25], forecasting protein solubility[26], modeling protein–protein interaction prediction[27], and exploring drug–target interactions[28]. Therefore, motivated by these successes in the use of GNNs, we introduce **HeteroTCR** (Fig. 1), a Heterogeneous Graph Neural Network based SPM that utilizes only CDR3β sequence information from TCRs and peptides to improve predictive accuracy for peptide-TCR recognition across various datasets. Firstly, we extract the max-pooling layers of the pre-trained Convolutional Neural Network (CNN) module as numeric embeddings of TCRs and peptides to input the model. Then, we employ Heterogeneous GNN module as the backbone of our model, allowing HeteroTCR to extract information on between-type (peptide-TCR) interaction and within-type (TCR-TCR or peptide-peptide)

similarity from input embeddings, thereby learning the probability of binding for classification. Existing SPMs mainly rely on the interaction information extracted from first-order neighborhoods while omitting insights originating from higher-order neighborhoods. An inherent advantage for HeteroTCR is that it can integrate information from neighborhoods of different orders, including both interaction information and similarity information. Our evaluation demonstrates that HeteroTCR outperforms state-of-the-art models on multiple independent datasets in terms of area under the curve (AUC) of the receiver operating characteristic (ROC). Ablation studies and visualization substantiate the essential role played by the Heterogeneous GNN module in enhancing the performance of HeteroTCR. This module adeptly captures critical features that underlie the binding process, showcasing its capacity for generalization across datasets with varying spatial distributions. Furthermore, we demonstrate HeteroTCR's robustness and reliability in single-cell datasets by correlating predicted probabilities with experimentally derived pMHC-T cell binding fractions.

## Results

### Model architecture

Conceptually, HeteroTCR divides the prediction of peptide-TCR interactions into three steps (Fig. 1). Firstly, a pre-training strategy is adopted, wherein the encoding vectors of the max-pooling layers before classification in CNN module are extracted as numeric embeddings of TCRs and peptides. Secondly, Heterogeneous GNN module is utilized to extract information on between-type (peptide-TCR) interaction and within-type (TCR-TCR or peptide-peptide) similarity. This results in TCR vectors containing information about interacting peptides and similar TCRs, and peptide

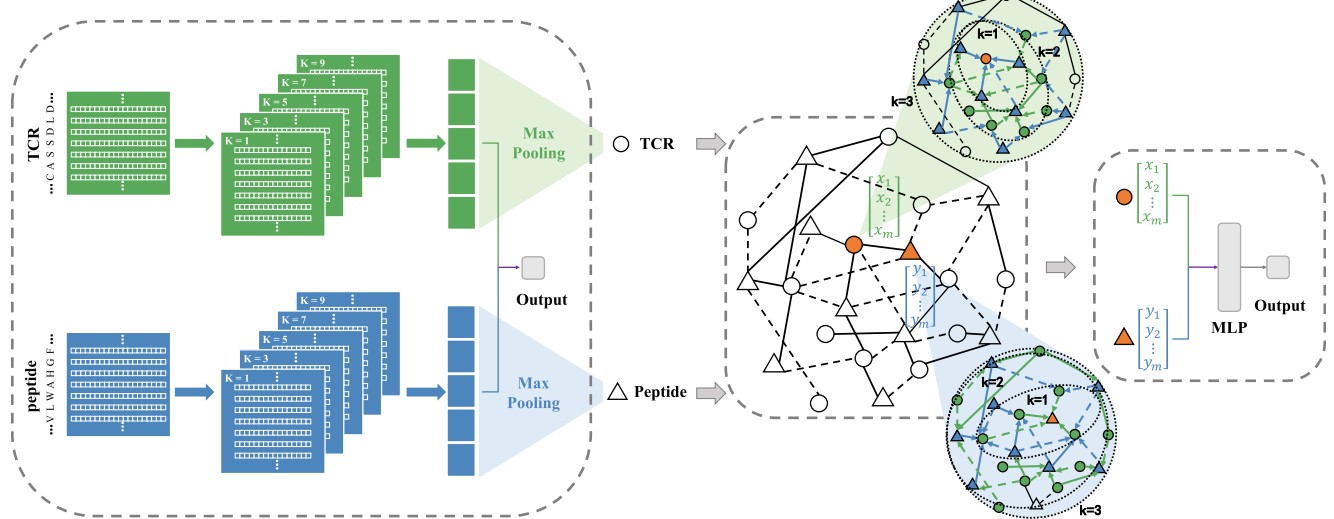

**Fig. 1 | Architecture of HeteroTCR.** To begin, the amino acid sequences of TCRs and peptides are encoded into numeric embeddings by a pre-trained CNN module. Subsequently, the Heterogeneous GNN module is employed to capture information pertaining to interactions between different types (peptide-TCR) and similarities within the same type (TCR-TCR or peptide-peptide). A circle node represents a TCR, while a triangle node represents a peptide. The edge of the solid line represents an interaction between a TCR and a peptide, while the dotted line represents no interaction, and no connection represents unknown. It is important to note that when initializing the graph, the nature of the relationship between peptides and TCRs is unknown, and we can only rely on the input training pairs to identify the existence of a relationship. We get the weight matrix by iterating 1000 epochs of training to learn and adjust the nature of the relationship between the given TCR and peptide, whether it is a solid line or a dotted line. We use a set of aggregator functions to learn aggregate feature information from $K^{th}$-order neighborhoods of a node, with a default K value of 3. As shown in the graphs with green and blue shading, $k = 1$, $k = 2$, and $k = 3$ represent the node information aggregated from the first, second, and third-order neighborhoods of the circle and triangle nodes. In other word, a

given peptide/TCR can learn information from its 1st-order binding TCRs/peptides, while the 1st-order TCR/peptide can learn information from its 1st-order binding peptides/TCRs which are 2nd-order neighborhoods of given peptide/TCR. Therefore, a given peptide/TCR can extract between-type (peptide-TCR) interaction information from the 1st- and 3rd-order TCRs/peptides, and obtain within-type (TCR-TCR or peptide-peptide) similarity information from the 2nd-order peptides/TCRs. This is because TCRs/peptides with shared features tend to interact with the same peptide/TCR. This process continues until order K is reached. Finally, an MLP is constructed utilizing these two numeric vectors to ascertain the presence of an interaction between a TCR and a peptide. Overall, HeteroTCR is composed of three main components: a pre-trained CNN, a Heterogeneous GNN, and an MLP classifier. The numeric embeddings extracted from the pre-trained CNN module are utilized as inputs for the Heterogeneous GNN module, which enables the extraction of information from the sequences. Specifically, the Heterogeneous GNN initially creates a global graph based on the entire dataset, and then trains on each pair of peptide-TCR inputs.

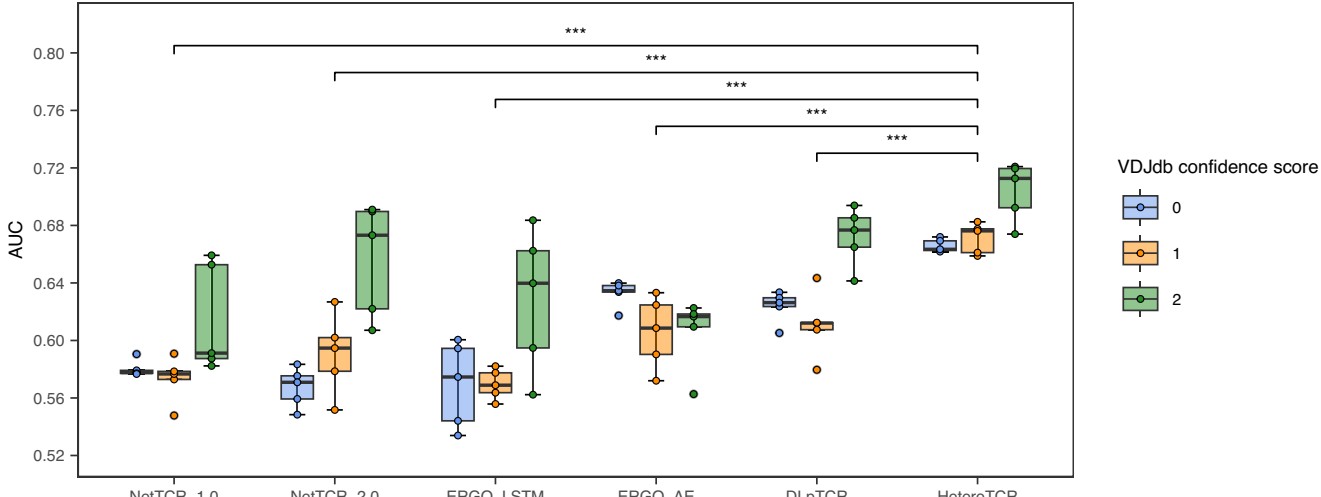

**Fig. 2 | Performance comparison between HeteroTCR and published methods based on pair-based data sets with 5-fold cross-validation.** All boxplots: The center line of each boxplot marks the sample median, and the box extends from the lower to the upper quartile. Each colored point represents one data point. Each boxplot represents the results of n = 5 independent experiments. Our results show that HeteroTCR outperforms tested methods, as indicated by a paired t-test P-value of <0.00001 for comparisons with any other model.

vectors containing information about interacting TCRs and similar peptides. Finally, a Multi-Layer Perceptron (MLP) is created on top of these two numeric vectors to identify whether an interaction exists between a TCR and a peptide.

To numerically embed TCR and peptide sequences, we adopt a pre-training strategy inspired by Montemurro et al[5] who propose that the ability of classification in CNN-based prediction models is driven by the representation in the max-pooled CNN layer. Therefore, we extract the numeric embeddings of TCRs and peptides from the max-pooled CNN layer of the pre-trained CNN module for subsequent module. First, we encode amino acid (AA) sequences of TCRs and peptides using the BLOSUM50 matrix[29], representing each AA as the score for substituting it with all 20 AAs. Then, TCR sequences and peptide sequences are individually processed using different convolution filters with kernel sizes {1, 3, 5, 7, 9}. Each kernel size's convolutional output is max-pooled, yielding a single vector with 160 entries (80 for each input sequence) concatenated from multiple feature vectors. This 160-dimensional vector is fed into an MLP, producing the probability of peptide-TCR binding (Methods). Finally, the max-pooling layers of the pre-trained CNN module are extracted as numeric embeddings of TCRs and peptides, ready for the subsequent module.

For interaction information and similarity information on TCRs and peptides, existing algorithms for predicting peptide-TCR interactions typically extract sequence information from peptides and TCRs separately and simply concatenate them for classification prediction. However, this approach may not directly focus on the interaction problem, but learn an embedding for the individual interactors, resulting in reduced model performance[23]. Moreover, a single TCR can bind to multiple peptides, and a single peptide can bind to numerous TCRs. TCRs (peptides) associated with the same peptide (TCR) may have similarities to some extent. These established conditions consequently enable the capture of within-type (TCR-TCR or peptide-peptide) similarity using sequence information. Therefore, we adopt Heterogeneous GNN, a feature extraction module that stores information about different types of entities in nodes and their different types of relations in edges, to capture the within-type similarity information. We consider the complete dataset as a global graph, featuring two node types: TCR and peptide, and two edge types: TCR binding to peptide and peptide binding to TCR. It is worth noting that the two edge types are set to be equal, ensuring a balanced interaction between TCRs and peptides.

The features of nodes in Heterogeneous GNN are numeric embeddings of TCRs and peptides obtained from the max-pooling layers of the pre-trained CNN module. A solid line represents an interaction between a TCR and a peptide, while the dotted line represents no interaction, and no connection represents an unknown status (Fig. 1). Messages are transmitted on the edges through the network, and the weight matrix is obtained through backpropagation from the final MLP classifier (Methods). We use a set of aggregator functions to learn aggregate feature information from $K^{th}$-order neighborhoods of a node, with a default K value of 3. In other words, a given peptide/TCR can learn information from its 1st-order binding TCRs/peptides, while the 1st-order TCR/peptide can learn information from its 1st-order binding peptides/TCRs which are 2nd-order neighborhoods of given peptide/TCR. Therefore, a given peptide/TCR can extract between-type (peptide-TCR) interaction information from the 1st- and 3rd-order TCRs/peptides, and obtain within-type (TCR-TCR or peptide-peptide) similarity information from the 2nd-order peptides/TCRs. This is because TCRs/peptides with shared features tends to interact with the same peptide/TCR. This process continues until order K is reached. Therefore, Heterogeneous GNN can extract information on between-type (peptide-TCR) interaction and within-type (TCR-TCR or peptide-peptide) similarity, allowing the vectors to learn about the probability of a binding event.

Finally, we utilize the numeric vectors of TCRs and peptides to learn their pairing. We construct an MLP (Methods) based on the Heterogeneous GNN module's outputs. The final MLP layer is a single neuron for predicting the interaction between a TCR and a peptide. HeteroTCR outputs a continuous variable between 0 and 1, reflecting the predicted binding strength. A value greater than or equal to 0.5 indicates that there is an interaction between a peptide and a TCR, while a value below 0.5 predicts no interaction.

## Comparisons with published methods on the independent testing dataset

We assessed the generalizability of HeteroTCR and other published methods on pair-based data sets (Methods). Specifically, the five models under comparison include NetTCR-1.0, NetTCR-2.0, ERGO LSTM, ERGO AE, and DLpTCR. NetTCR-1.0[20] is based on a simple 1D convolutional neural network (CNN), integrating peptide and CDR3$\beta$ sequence information encoded with BLOSUM50 matrix for the prediction of peptide-TCR specificity. NetTCR-2.0[5] modifies the network structure on the basis of NetTCR-1.0 and uses paired CDR3$\alpha$/$\beta$ data as input instead of CDR3$\beta$ information only. Here, we exclusively employed the CDR3$\beta$ single-input model from NetTCR-2.0 for a fair comparison, omitting the utilization of the paired CDR3$\alpha$/$\beta$ dual-input model. ERGO[21] applies two encoding

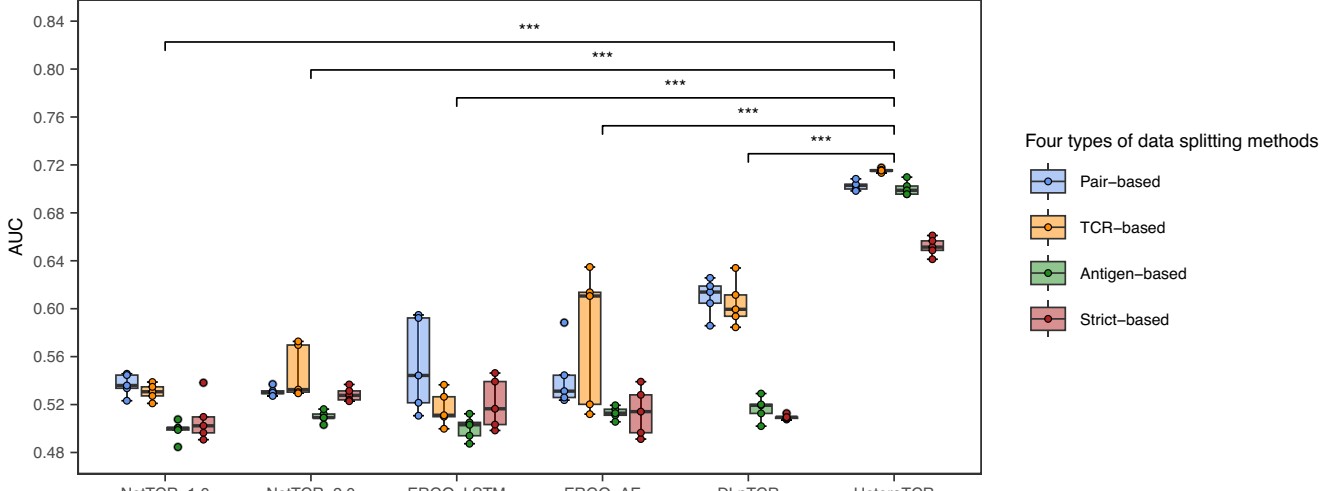

**Fig. 3 | Performance comparison with HeteroTCR and published methods based on four types of data splitting methods with VDJdb confidence score greater than or equal to 0.** For pair-based data sets, the validation dataset McPAS-TCR consists of 1574 pairs, and the testing dataset consists of 9206 pairs. For TCR-based data sets, antigen-based data sets, and strict-based data sets, the number of pairs in McPAS-TCR is 1436, 836, and 784, respectively, while VDJdb contains 7374, 1600, and 1208 pairs, respectively. All boxplots: The center line of each boxplot marks the sample median, the colored points scattered along each boxplot represent all the actual data points, and the box extends from the lower to upper quartile. Each boxplot represents the results of $n = 5$ independent experiments. The paired t-test P-value between the AUC of HeteroTCR and that of any other model is < 0.00001, indicating that HeteroTCR performs significantly better than other methods.

methods, Long Short Term Memory (LSTM) acceptor encoding, and Autoencoder-based encoding, for predicting TCR-peptide binding. DLpTCR[22] uses a variety of mixed encoding methods, including one-hot, PCP, and PCA encoding, and integrates FCN, LeNet-5 and ResNet-20 to predict the interaction between peptide and TCR.

The codes for the aforementioned models is sourced from their respective GitHub repositories. The models were retrained and evaluated on our dataset. We trained the models on the IEDB dataset[30], which comprises of 76,348 peptide-TCR pairs, using 5-fold cross-validation. The dataset was randomly divided into five equal parts, and the models were evaluated in each part in a rotating manner (Methods). For independent testing, we used data collected from the VDJdb[31], which contains peptide-TCR pairs with confidence scores ranging from 0 to 3, where higher scores indicate more reliable records. Peptide-TCR pairs present in the training data were excluded from the testing data. Throughout our analysis, the area under the curve (AUC) of the receiver operating characteristic (ROC) was used as the metric to estimate model performance. All experiments were performed with default parameter settings.

As shown in Fig. 2, the AUC scores of all models generally increased with improved dataset quality. However, due to a limited number of data pairs in VDJdb with a confidence score greater than or equal to 3 (only 32 pairs), yielding statistically significant results became impractical, leading us to exclude them. Additionally, the AUCs of HeteroTCR reached 0.6658, 0.6712, and 0.7039 in VDJdb with a confidence score greater than or equal to 0, 1, and 2 respectively, surpassing those of other published models (paired t-test P-value between HeteroTCR's AUC and that of any other model was <0.00001) (details in Supplementary Tables 1, 2).

**Four types of data splitting methods to evaluate the generalization ability of the model**

To further demonstrate the generalizability of HeteroTCR, we evaluated models based on four different types of data splitting methods (Methods), namely pair-based, TCR-based, antigen-based, and strict-based data sets. For the pair-based data sets, the peptide-TCR pairs present in the training dataset were removed from the testing dataset. For TCR-based data sets, peptide-TCR pairs involving TCRs present in the training dataset are excluded from the testing dataset. Similarly, for antigen-based data sets, peptide-TCR pairs involving peptides appearing in the training dataset are excluded from the testing dataset. In the case of strict-based data sets,

peptide-TCR pairs involving either TCRs or peptides present in the training dataset are excluded from the testing dataset.

In this analysis, we trained models on IEDB data using four types of data splitting methods, and took McPAS-TCR[32] as the validation dataset for parameter selection. The evaluation was conducted on VDJdb with varying confidence scores (Methods). Importantly, the validation dataset was completely independent from the training set, and likewise, the testing dataset was also independent from both training and validation sets. This approach confirmed the model's grasp of essential peptide-TCR binding properties, rather than confounding factors within the database.

As shown in Fig. 3 (performance comparison with other VDJdb confidence scores are detailed in Supplementary Figs. 1–3), we observed a general decrease in AUC scores across all models as data splitting methods became stricter. This decline in performance could stem from potential violations of the assumption of independent and identically distributed (i.i.d.) data across training, validation, and testing sets[7]. Additionally, stricter data splitting methods might lead to weaker data leakage and consequent performance degradation. Given the sparsity of current datasets, it is challenging for models to generalize to unseen epitopes. Notably, all models exhibited substantial performance drops on antigen-based and strict-based data sets as anticipated.

HeteroTCR significantly outperformed other models across all dataset splitting with varying VDJdb confidence scores (paired t-test P-value between the AUC of HeteroTCR and that of any other model was <0.00001) (see Supplementary Tables 3, 4 for more details). Although generalizability on strict-based data sets remained limited, HeteroTCR's performance (AUC = 0.6535) surpassed random guessing (AUC = 0.5). These findings indicated HeteroTCR's ability to learn from existing datasets and extrapolate predictions to unseen epitopes and TCRs.

Additionally, the lower variance in AUC for HeteroTCR compared to other methods was likely attributed to the robustness of the GNN model, which consistently performs well across diverse datasets and demonstrates a more stable predictive performance. This stability is a noteworthy characteristic that enhances the reliability and consistency of HeteroTCR's predictive capabilities.

**Comparisons to the state-of-the-art models**

ImRex[23] and TITAN[7] are two state-of-the-art (SOTA) models for predicting unseen epitopes and TCRs on independent datasets. ImRex relies on an

### Table 1 | Comparison of AUC between HeteroTCR and ImRex

| Settings | ImRex | HeteroTCR | Number of samples in training data (positive/negative) | Number of samples in testing data (positive/negative) |
|---|---|---|---|---|
| Shared-epitope data | 0.59 | **0.69** | 6702/6702 | 4101/4101 |
| Unique-epitope data | 0.54 | **0.61** | 6702/6702 | 736/736 |

Best performance is marked in bold. Data of ImRex is referenced from ref. 23. We generated the same number of negative samples for each dataset using the random shuffling approach.

### Table 2 | Comparison of AUC between HeteroTCR and TITAN

| Models | TCR split | | Strict split | |
|---|---|---|---|---|
| | Training data (positive/ negative) 9539/9539 | Testing data (positive/ negative) 1060/1060 | Training data (positive/ negative) 9171/9171 | Testing data (positive/ negative) 1008/1008 |
| TITAN K-NN | 0.79 ± 0.01 | | 0.54 ± 0.03 | |
| TITAN AA CDR3 | 0.75 ± 0.02 | | 0.60 ± 0.04 | |
| TITAN AA full | 0.76 ± 0.007 | | 0.59 ± 0.04 | |
| TITAN SMI CDR3 | 0.73 ± 0.007 | | 0.60 ± 0.06 | |
| TITAN SMI full | 0.75 ± 0.006 | | 0.59 ± 0.06 | |
| TITAN Pretrained | 0.81 ± 0.01 | | 0.56 ± 0.04 | |
| TITAN Pretrained aug. | 0.80 ± 0.01 | | 0.59 ± 0.03 | |
| TITAN Pretrained semifrozen | 0.80 ± 0.01 | | 0.58 ± 0.06 | |
| TITAN Pretrained semifrozen aug. | 0.82 ± 0.01 | | 0.62 ± 0.06 | |
| HeteroTCR | **0.83** ± 0.01 | | **0.67** ± 0.09 | |

Data of TITAN is referenced from ref. 7. Mean and standard deviation of each model configuration on TCR split and strict split. The best performance is marked in bold. K-NN refers to the baseline model of TITAN. AA means sequences are encoded by amino acid while SMI refers to peptides that are encoded by SMILES. CDR3 denotes only CDR3 sequences of TCRs are input of the model while full denotes full sequences of TCRs are fed to the model. All TITAN Pretrained models adopt SMILES encodings of peptides and full sequence input for TCRs. Pretrained means the model is pre-trained on BindingDB, aug means the model is pre-trained on BindingDB with data augmentation, and semifrozen means the weights in the epitope channel are fixed during fine-tuning.

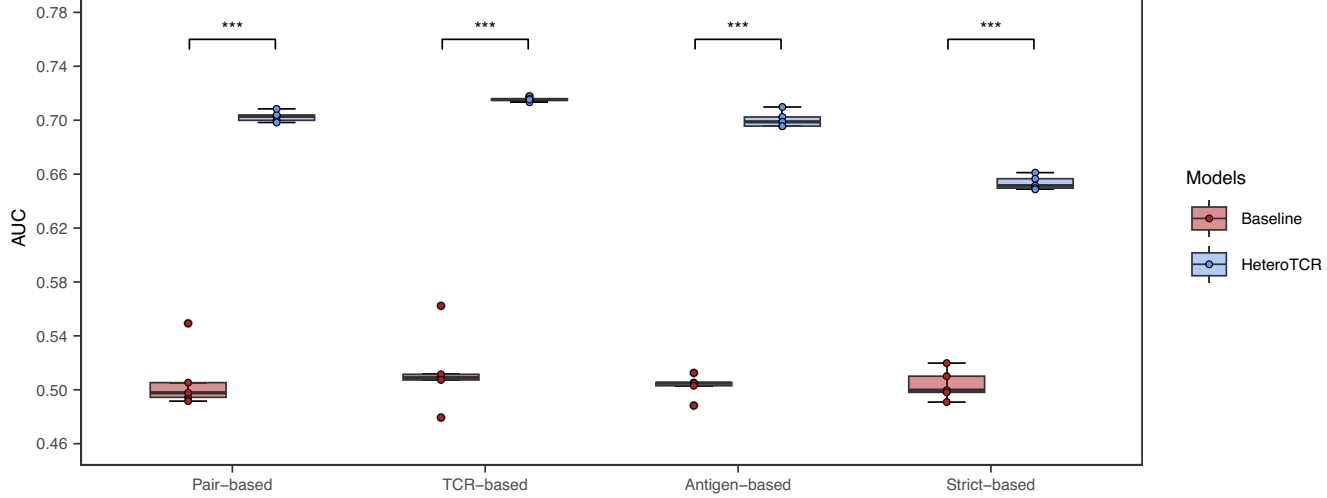

**Fig. 4 | Performance comparison with HeteroTCR and the baseline model based on four types of data splitting methods with VDJdb confidence score greater than or equal to 0.** All boxplots: The center line of each boxplot marks the sample median, the colored points scattered along each boxplot represent all the actual data points, and the box extends from the lower to upper quartile. Each boxplot represents the results of $n$ = 5 independent experiments. T-test P-value between HeteroTCR and the baseline model is <0.001.

interaction map, which combines the physicochemical properties of both interactors on the amino-acid level, to predict peptide-TCR recognition under different dataset settings. TITAN exploits convolutions with an interpretable attention mechanism to aggregate local information and integrates the modalities, from which binding probabilities are predicted. To compare HeteroTCR directly, we trained and tested it under their respective training and testing data.

In ImRex (Table 1), the VDJdb dataset was used for training with 5-fold cross-validation, and the McPAS-TCR dataset served as an independent testing set, which was split into two subsets: one containing peptides already present in the training set (shared-epitope data), and the other containing peptides not seen during ImRex training (unseen-epitope data, which was referred as the unique-epitope data by the authors of ImRex[23]). Negative interaction pairs were generated using a shuffling approach for each subset.

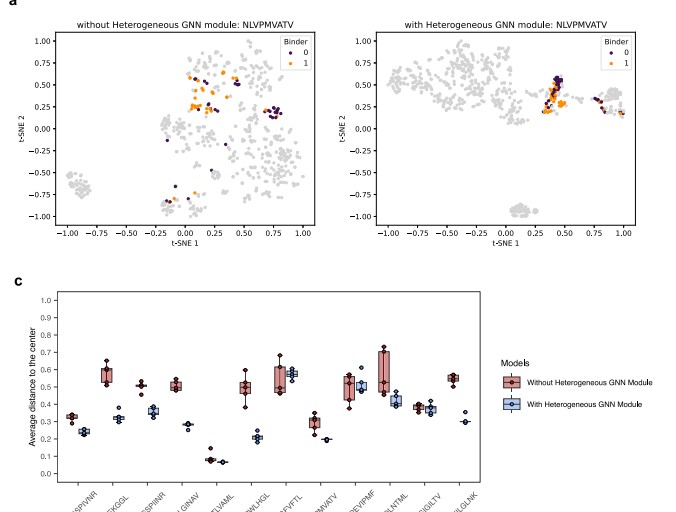
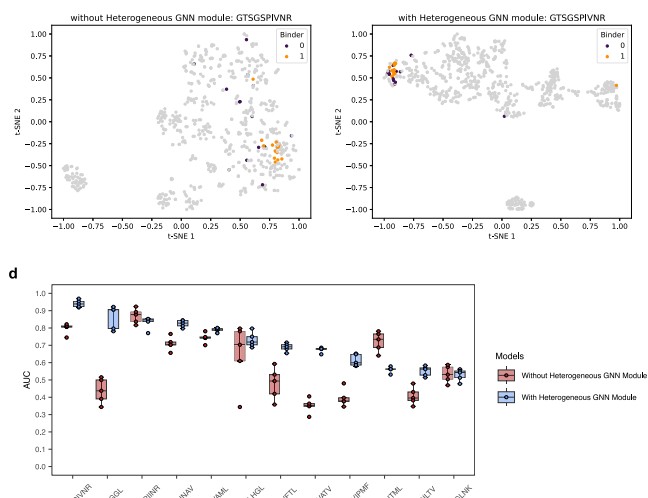

**Fig. 5 | Superiority and interpretability of the design of HeteroTCR. a, b** The t-SNE plot for the peptide-TCR pairs of the given peptide with and without a Heterogeneous GNN module. Each point represents a pair of peptide-TCR and is colored purple or orange, where purple represents no interaction and orange represents an interaction. Without Heterogeneous GNN module, the colored dots are scattered, and no clear boundary can be observed between purple and orange dots. In contrast, with Heterogeneous GNN, the colored dots are more clearly clustered and classified. **c** The degree of aggregation of the colored points (peptides with ≥15 cognate TCRs) is assessed by computing the center of the point set and the average distance of each point to the center. All boxplots: The center line of each

boxplot marks the sample median, the colored points scattered along each boxplot represent all the actual data points, and the box extends from the lower to upper quartile. Each boxplot represents the results of $n = 5$ independent experiments. **d** The per-peptide (with ≥15 cognate TCRs) AUC for the models with and without Heterogeneous GNN module, which are trained five times on IEDB and tested on VDJdb with confidence score greater than or equal to 1, based on the pair-based setting. All boxplots: The center line of each boxplot marks the sample median, the colored points scattered along each boxplot represent all the actual data points, and the box extends from the lower to upper quartile. Each boxplot represents the results of $n = 5$ independent experiments.

We observed that HeteroTCR consistently outperformed ImRex on both independent testing sets.

For comparison with TITAN (Table 2), two different data splitting methods were utilized to evaluate model generalizability using 10-fold cross-validation. The TCR split ensured each TCR appeared in only one fold, avoiding overlap between validation and training datasets. The strict split extended this approach to both TCRs and peptides, ensuring that validation data were unseen during training. To ensure the separation of TCRs and peptides in their folds, negative data were generated by shuffling within each fold.

We found that HeteroTCR, utilizing only CDR3 sequences of TCRs and AA sequences of peptides, outperformed TITAN's best model, which was pretrained on BindingDB[33] and fine-tuned on training data with full TCR sequences and SMILES encodings[34] of peptides, featuring a frozen epitope input channel with augmentations to enrich the data. HeteroTCR exhibited improved performance on unseen epitopes and TCRs with the strict data-splitting method.

**The Heterogeneous GNN module is essential for the improved performance of HeteroTCR**

To emphasize the advantages of HeteroTCR, we conducted ablation studies by establishing a baseline model. In computational terms, ablation studies refer to the systematic removal of components from the computational model to assess its impact on overall performance, rather than modifying predicted residues of peptides or TCRs. In our baseline model, we excluded the Heterogeneous GNN module, enabling us to assess the module's contribution to enhancing both between-type (peptide-TCR) interaction and within-type (TCR-TCR or peptide-peptide) similarity. Models were trained on IEDB data using four types of data splitting methods, with McPAS-TCR as the validation dataset. Further evaluation took place independently on VDJdb with a confidence score greater than or equal to 0.

The AUC of HeteroTCR markly surpassed that of the baseline model (Fig. 4), indicating that the information on between-type interaction and within-type similarity learned by Heterogeneous GNN greatly enhanced the

model's performance (further details can be found in Supplementary Tables 5 and 6).

Furthermore, we investigated the relationship between model performance and the parameter K (Supplementary Table 7). We considered K values ranging from 1 to 6. Our findings indicate a non-linear relationship between K and model performance, characterized by an initial improvement followed by a subsequent decline. Notably, the model achieves the highest AUC at $K = 4$. However, considering the associated increase in training depth and computational cost at higher K values, we recommend adopting $K = 3$ as it provides a balanced trade-off between time complexity and performance.

An inherent challenge in generic peptide-TCR models is their potential to memorize TCR motifs regardless of the peptide partner, leading to an inability to capture the true underlying molecular forces governing the binding process and merely managing to memorize spurious motifs present within the TCR training data[23]. To assess whether HeteroTCR captures essential binding features or memorizes spurious TCR motifs, we visualized model-extracted representations of peptide-TCR pairs in a 2-dimensional space using t-distributed stochastic neighbor embedding (t-SNE)[35] (Fig. 5a, b). Due to the large number of peptide-TCR pairs in VDJdb with a confidence score greater than or equal to 0, we selected representative visuals from VDJdb with a confidence score greater than or equal to 1. Notably, using numeric representations extracted by the full model, cognate TCRs for a specific antigen clustered more tightly compared to a model without the Heterogeneous GNN module. Additionally, with the Heterogeneous GNN module, we observed a better separation between the orange dots (representing positive samples, indicating peptide-TCR interactions) and the purple dots (indicating negative samples, denoting no peptide-TCR interactions).

To measure the degree of aggregation of positive and negative samples for the same antigen, we calculated the center of the point set and the average distance of each point to the center (Fig. 5c). We found that the degree of aggregation of the samples with the Heterogeneous GNN module, is smaller than that without the Heterogeneous GNN module, indicating that cognate

**a**

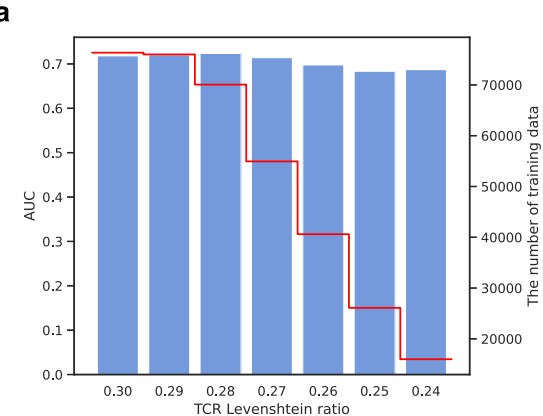
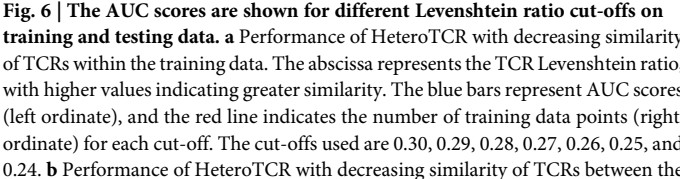

**b**

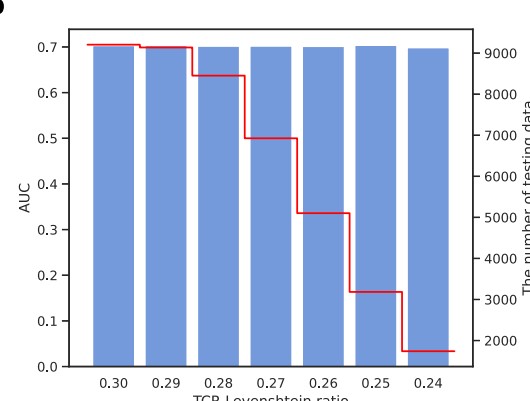

**Fig. 6 | The AUC scores are shown for different Levenshtein ratio cut-offs on training and testing data. a** Performance of HeteroTCR with decreasing similarity of TCRs within the training data. The abscissa represents the TCR Levenshtein ratio, with higher values indicating greater similarity. The blue bars represent AUC scores (left ordinate), and the red line indicates the number of training data points (right ordinate) for each cut-off. The cut-offs used are 0.30, 0.29, 0.28, 0.27, 0.26, 0.25, and 0.24. **b** Performance of HeteroTCR with decreasing similarity of TCRs between the

training and testing data. The abscissa represents the TCR Levenshtein ratio, where higher values indicate that testing TCRs are more similar to the training TCRs. The blue bars represent AUC scores (left ordinate), and the red line indicates the number of testing data points (right ordinate) for each cut-off. The cut-offs used are 0.30, 0.29, 0.28, 0.27, 0.26, 0.25, and 0.24, corresponding to thresholds where similar TCR data points are progressively removed from the testing dataset.

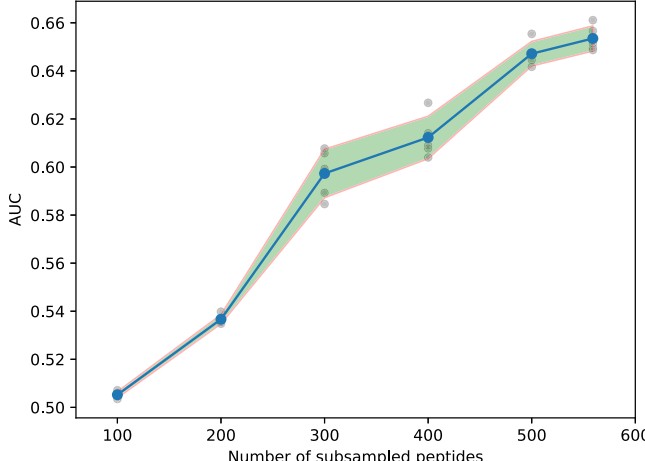

**Fig. 7 | Performance of HeteroTCR across different number of subsampled peptides based on strict-based data sets.** The shaded area depicts the mean ± the standard deviation over five repeated experiments. The blue points represent the mean data points, while the grey points ($n = 5$) scattered around each mean point represent all the actual data points. In the validation dataset and testing dataset, the number of pairs for different numbers of subsampled peptides is 784 and 1208, respectively. For different numbers of subsampled peptides (ranging from 100 to 559), the corresponding numbers of pairs in the training dataset are 7180, 22440, 40932, 52636, 69816, 76348, respectively.

TCRs for a specific antigen cluster more tightly (paired t-test P-value between the models with and without Heterogeneous GNN module is $3.9 \times 10^{-10}$) (additional details in Supplementary Table 8). Figure 5d shows the AUC in per-peptide classification of positive and negative samples (≥15 cognate TCRs), demonstrating the Heterogeneous GNN module's reasonable performance (paired t-test P-value between the models with and without Heterogeneous GNN module is $1.2 \times 10^{-7}$) (additional details in Supplementary Table 9). This observation reveals that the information on between-type (peptide-TCR) interactions and within-type (TCR-TCR or peptide-peptide) similarity learned by HeteroTCR prevents reliance on TCR motif patterns for interaction prediction.

TCRs recognizing the same antigen can exhibit diverse amino acid sequences. To investigate the impact of TCR sequence diversity on HeteroTCR, we evaluated model performance using TCRs with decreasing

similarity within the training data (Fig. 6a), and trained with the full data but tested using TCRs with decreasing similarity to the training data (Fig. 6b). TCR similarity was measured using the mean Levenshtein ratio for each TCR to all TCRs in the training data (Methods). A higher mean Levenshtein ratio suggests a TCR sequence is closer to the TCRs in the training data. AUC analysis was conducted for TCRs below specified Levenshtein ratio cut-offs (Methods). AUC scores are shown for model trained (Fig. 6a) or tested (Fig. 6b) with TCRs above each cut-off. The outcomes reveal HeteroTCR's relative robust as TCR similarity levels decrease. With diminishing TCR similarity within the training set, HeteroTCR's performance slightly drops (Fig. 6a), indicating a subtle influence of TCR motifs on the model. As the similarity between testing and the training TCRs (Fig. 6b) wanes, AUC scores remain steady, indicating HeteroTCR's capability to generalize across diverse TCR motifs.

Additionally, we investigated the impact of the number of peptides on model performance (Fig. 7). In this analysis, we trained models on the IEDB dataset using strict-based datasets and utilized McPAS-TCR as the validation dataset for parameter selection. The evaluation was performed on VDJdb with a confidence score greater than or equal to 0. In the experiments, the initial training dataset consisted of 559 unique peptides. We randomly subsampled them to reduce the quantity to 100, 200, 300, 400, and 500. Our observations revealed that the model performance continued to improve with increased input data, suggesting that HeteroTCR benefited from the inclusion of more peptides (additional details in Supplementary Table 10). In summary, our experiments demonstrate that increasing the number of peptides in the training dataset consistently improves the performance of the model, underscoring the importance of diverse antigenic exposures in enhancing predictive accuracy.

### HeteroTCR binding predictions correlate with experimentally derived pMHC-T cell binding fractions

To assess the performance of the model from an alternative angle, we investigated whether the predictions of the HeteroTCR model correlate with experimentally derived pMHC-T cell binding fractions (Methods). The data utilized in our study was generated from the 10x Genomics Chromium Single Cell Immune Profiling platform[36] (Methods). We analyzed single-cell datasets containing profiles of CD8[+] T cells specific to 44 different pMHC multimers, sourced from four healthy donors. The binding specificity between each T-cell and tested pMHC was quantified by counting the number of unique molecular identifier (UMI) sequences associated with each specific pMHC in the T-cell. After data curation and the computation

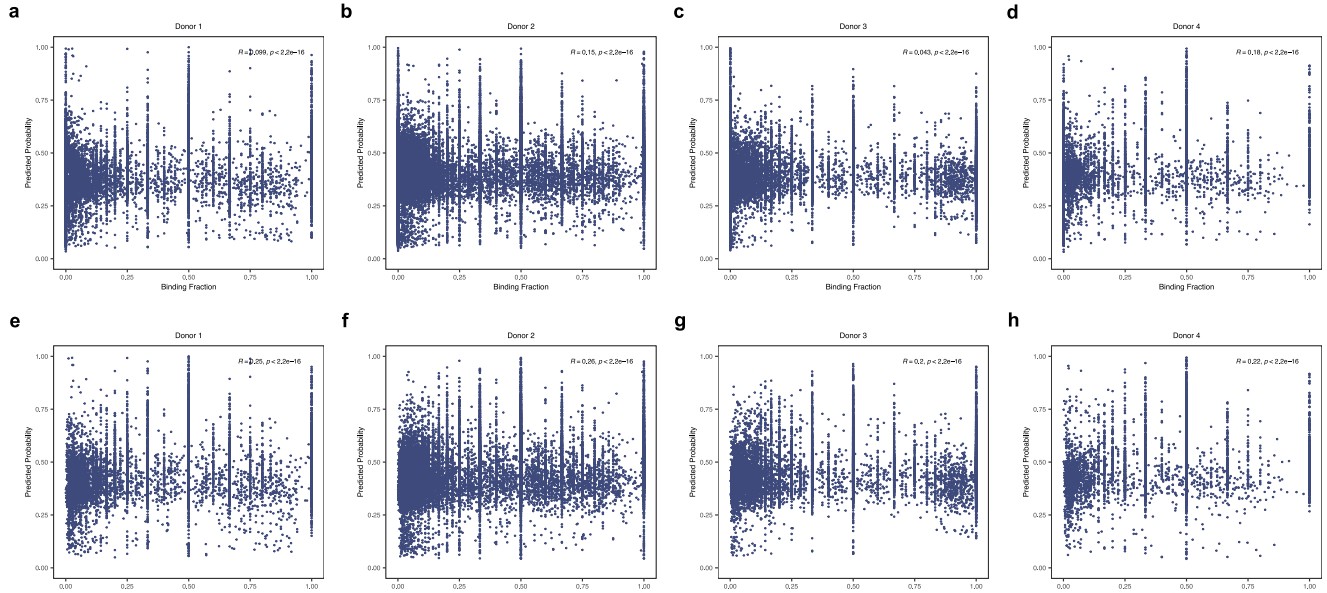

**Fig. 8 | Predicted binding probability is positively correlated with binding fractions. a–d** Binding predictions for donors 1–4 correlate with binding fractions, where binding fractions include cases equal to 0. The number of data for each donor is 29495, 62956, 41496, and 13430, respectively. **e–h** Binding predictions for donors 1–4 correlate with binding fractions, where binding fractions exclude cases equal to 0. The number of data for each donor is 9909, 24156, 13114, and 4792, respectively.

of binding fractions (Methods), we proceeded to calculate the Spearman correlation coefficient between HeteroTCR-predicted binding probabilities and binding fractions, i.e. the fraction of a T cell clone bound to a specific TCR.

Figure 8a–d illustrates the correlation between the predicted binding probabilities of HeteroTCR and binding fractions. Notably, the dataset includes cases where the binding fraction is 0, illustrating an absence of binding specificity. A positive correlation observed in the figures indicates the association between the model's predicted probabilities and the binding events. In Fig. 8e–h, the correlation between the two variables in data excluding instances where the binding fraction equals 0. Consequently, the figures exclusively comprise data with binding specificity, and a positive correlation depicts that the predicted binding probabilities of HeteroTCR are associated with binding strength in instances where binding occurs.

## Discussion

In this study, we introduce HeteroTCR, a Heterogeneous Graph Neural Network method leveraging the sequence information form the peptide and CDR3$\beta$ region of TCR for predicting peptide-TCR binding probability. HeteroTCR is designed to extract both information on between-type (peptide-TCR) interaction and within-type (TCR-TCR or peptide-peptide) similarity from input sequence. As far as we know, HeteroTCR is the first model to apply graph neural networks to peptide-TCR interaction prediction tasks. Our evaluation demonstrated that HeteroTCR outperforms existing methods in predicting peptide-TCR binding probabilities, particularly across various independent testing datasets and dataset settings. Although the generalization capability of HeteroTCR on strict-based data sets is limited, its overall performance remains robust. These results highlight HeteroTCR's enhanced reliability and robustness in predicting immunogenic peptides recognized by TCR. The HeteroTCR program, trained model parameters, machine learning platform and hardware (Methods), and datasets are available on GitHub (https://github.com/yuzilan/HeteroTCR).

To further analyze the superiority of HeteroTCR design, we establish a baseline model for ablation studies. Our results show that the information on between-type interaction and within-type similarity learned by the Heterogeneous GNN module enhances model performance considerably. Utilizing t-SNE visualization on numeric vectors processed with and without the Heterogeneous GNN module, we uncover HeteroTCR's ability to capture fundamental molecular binding forces rather than memorizing TCR pseudo motifs. Additionally, we investigate the impact of TCR pattern similarity on HeteroTCR by varying Levenshtein ratio cut-offs on training and testing sets, illustrating that HeteroTCR's strong performance is partly contributed by within-type similarity.

Moreover, we investigated the correlation between HeteroTCR's binding predictions and experimentally derived pMHC-T cell binding fractions using publicly available 10x Genomics data from single cell droplet sequences in the presence of DNA barcoded pMHC dextramers. While the model is promising in aiding the identification of immunogenic epitopes, a wide range of binding strengths were observed even among TCR with the highest HeteroTCR GNN-inferred interaction scores. This underscores that HeteroTCR may have the utility to prioritize candidate TCRs of clinical relevance, but binding prediction will require experimental validation.

Heterogeneous GNN, integrating diverse node and edge information[37], opens promising avenues for future research to expand on HeteroTCR. Additional information, including the alpha chains of TCRs, *V/J* genes, other CDR loops, and the MHC contexts, can be incorporated into the graph as distinct node types with different vector dimensions and embedding methods. Edges can also have weights attached to represent different degrees of importance and the proportion of message transmission. Notably, HeteroTCR has important implications for designing TCR-T or neoantigen vaccine therapies, and offers a modeling framework for various molecular interactions like protein-protein or antibody-antigen interactions.

## Methods
### Data

Here, we collected data from three well-known databases: IEDB[30], VDJdb[31], and McPAS-TCR[32]. The Immune Epitope Database (IEDB) is a freely available resource that contains a comprehensive collection of antibody and T cell epitopes in humans, non-human primates, and other animal species in the context of infectious disease, allergy, autoimmunity, and transplantation. The IEDB dataset was downloaded from http://www.iedb.org on November 17, 2021. The VDJdb is a curated database of TCR sequences with known antigen specificities and confidence scores to highlight the most reliable records. The McPAS-TCR is a manually curated catalog that links TCR sequences to their antigen targets or to the pathology and organ with

which they are associated. The VDJdb and McPAS-TCR were collected from https://vdjdb.cdr3.net and http://friedmanlab.weizmann.ac.il/McPAS-TCR/, both on June 12, 2021. To construct reliable datasets, we took five preprocessing steps on the raw data.

Step 1. filtering data: (i) The data used in this study were restricted to TCR$\beta$ chain sequences due to the scarcity of paired chain data. (ii) Peptide-CDR3$\beta$ pairs were adopted because the complementarity-determining region 3 (CDR3) is the key determinant of specificity in antigen recognition. (iii) Given that peptide with lengths ranging from 8 to 15 amino acids (AAs) and CDR3$\beta$ sequences containing 10–20 AAs account for the majority of datasets[22], we focused on the CDR3$\beta$ sequences with 10–20 AAs and peptides with 8–15 AAs presented by human MHC class I molecules, as they are likely to represent the most common and biologically relevant cases.

Step 2. removing invalid sequences: Since the majority of CDR3$\beta$ chains begin with the amino acid 'C' and end with 'F'[38], we excluded peptide-CDR3$\beta$ pairs whose CDR3$\beta$ sequences do not adhere to this pattern. Furthermore, we eliminated peptide-CDR3$\beta$ sequences containing lowercase letters or letters that are not amino acids as these likely result from errors during database collection and collation.

Step 3. removing redundant sequence pairs: Duplicate sequence pairs may exist in the databases due to the omission of information such as the CDR1 and CDR2 regions and HLA molecules. To address this, we removed redundant data within each database by retaining only unique peptide-CDR3$\beta$ sequence pairs.

Step 4. screening antigen-specific TCR CDR3s: iSMART[14] is a computational method that performs a specially parameterized pairwise local alignment on TCR CDR3 sequences to group them into antigen-specific clusters with high efficiency. Since TCRs usually share conserved sequence features when recognizing the same pMHC epitope[22], we employed iSMART as a filter to identify and cluster antigen-specific TCR CDR3s, removing the CDR3s that are not selected in the cluster.

Step 5. constructing datasets: The peptide-CDR3 pairs processed in Steps 1–4 can be added to the set of interacting pairs of CDR3s and peptides. However, to train supervised models, both positive and negative examples, which are pairs of CDR3s and peptides that do not recognize each other, are required. To generate negative data, each CDR3 sequence in the positive dataset was paired with a peptide that has not been shown to interact with the corresponding CDR3 by shuffling or mismatching the sequences. The enormous diversity of potential TCR and peptide sequences makes it unlikely for a randomly selected TCR to bind to a specific peptide, thereby making the shuffling method feasible[39]. In this study, the shuffling method was used instead of sampling uniformly from healthy CDR3 reference repertoire sets, as using an external negative reference dataset may introduce an inherent bias that may result in over-optimistic performance[23].

In the Results 2.2 section, we used IEDB dataset as the training dataset with 5-fold cross-validation to generate five optimal parameter models, and evaluated these five models on an independent testing dataset (VDJdb) to get five AUC metrics.

In the Results 2.3 section, we trained the model on the data from IEDB dataset, took McPAS-TCR as the validation dataset to select the optimal parameter model, and evaluated the model on VDJdb. We repeated this process five times, so we got five optimal parameter models each time and five AUC metrics on the independent testing dataset.

In the Results 2.4 section, to avoid being accused of manipulating data using complex filtering steps, we directly used the data of SOTA models to train and evaluate HeteroTCR and compared the results in their papers.

## Four types of data splitting methods

In order to further refine our experiment and show the performance of HeteroTCR, we redefined the data splitting methods into four types:
- Pair-based data sets/seen epitopes and seen TCRs.
- TCR-based data sets/unseen TCRs.
- Antigen-based data sets/unseen epitopes.
- Strict-based data sets/unseen epitopes and unseen TCRs.

For the pair-based data sets, it simply guarantees that the interacting peptide-TCR pairs in the testing dataset are not contained in the training dataset. It is worth noting that the peptide or TCR alone in the training dataset may exist in the testing dataset. This division method has been widely adopted by most models.

For the TCR-based data sets, if a TCR appears in the training dataset, it cannot be present in the testing dataset.

For the antigen-based data sets, the peptide that appears in the training dataset will no longer appear in the testing dataset.

For the strict-based data sets, it is necessary to ensure that both the TCR and the peptide in the training dataset do not appear in the testing dataset. This approach differs from the pair-based data sets, where the model's generalization ability is evaluated by predicting whether TCRs or peptides that are not in the database can stably bind.

In addition to the above four types of data splitting methods, our model utilized cross-validation across different datasets to demonstrate that our model learns essential information about TCRs and peptides, rather than simply memorizing the specifics of a single database. More details regarding the datasets can be found in the results section.

## Embedding of TCRs and peptides

Firstly, the TCR sequences were zero-padded to a maximum length of 20, while the peptide sequences to a maximum length of 15. We then encoded AA sequences of TCRs and peptides using the BLOSUM50 matrix, in which each AA is represented as the score for substituting the AA with all the 20 amino AAs. Hence, the BLOSUM encoding scheme maps a sequence with a length of $l$ to an array with a size of $l \times 20$. Secondly, the TCR sequences and the peptide sequences were separately deconvoluted by different convolution kernels with kernel size {1, 3, 5, 7, 9}, in which different features were integrated through different convolution kernels to filter the whole sequence. For each kernel size, the convolutional output was max-pooled and the resulting feature vectors were concatenated in a single vector with 160 entries (80 for each input sequence). Thirdly, the 160-dimensional vector was fed into a dense layer of 32 hidden neurons and the output consists of one single neuron, giving the probability of a peptide-TCR pair to bind (Fig. 1). The activation function used through the network is the sigmoid function. The model was trained for 1000 epochs with early stopping and patience of 50 epochs. The weights were updated using the Adam optimizer with a learning rate of 0.001. The batch size is 512 and the loss function is binary cross-entropy. Finally, the 160-dimensional max-pooling layers of the pre-trained CNN module were extracted as numeric embeddings of TCRs and peptides to input the next step.

## Heterogeneous GNN module

Firstly, we regard the entire dataset as a global graph. A circle node represents a TCR, while a triangle node represents a peptide (Fig. 1). The edge of the solid line (label 1) represents an interaction between a TCR and a peptide, while the dotted line (label 0) represents no interaction, and no connection represents unknown. The difference between solid and dotted lines is only used for the calculation of loss function and backpropagation of the MLP in the training dataset, as well as the evaluating the model in the testing dataset, and it does not participate in any other process to cause data leakage. In other words, when initializing the graph, the nature of the relationship between peptides and TCRs is unknown, and we can only rely on the input training pairs to identify the existence of a relationship. We get the weight matrix by iterating 1000 epochs of training to learn and adjust the nature of the relationship between the given TCR and peptide, whether it is a solid line or a dotted line.

Secondly, the idea behind Heterogeneous GNN can be broadly divided into two steps. We use a set of aggregator functions to learn aggregate feature information from $K^{th}$-order neighborhood of a node, with a default K value of 3. The activation function used through Heterogeneous GNN is the Leaky ReLUs function. The input of Heterogeneous GNN is an 80-dimensional numeric embedding of TCR and an 80-dimensional numeric embedding of

peptide, and the output is a 1024-dimensional TCR vector and a 1024-dimensional peptide vector.

**Notations.** Let $T = \{t_1, t_2, \ldots, t_m\}(|T| = m)$ and $P = \{p_1, p_2, \ldots, p_n\}$ $(|P| = n)$ denote the set of TCRs and peptides, respectively. Let $I^+ = \{y_{t,p} | t \in T, p \in P\}$ denote the positive interactions, where $y_{t,p}$ indicates that TCR $t$ has interacted with peptide $p$. Let $I^- = \{y_{t,p} | t \in T, p \in P\}$ denote the negative interactions, where $y_{t,p}$ indicates that TCR $t$ has no interacted with peptide $p$. Moreover, a peptide-TCR interaction graph is constructed, denoted as $\mathscr{G}(\mathscr{V}, \mathscr{E})$, where $\mathscr{V} = T \cup P$ is the set of nodes and $\mathscr{E} = \{(t, p) | y_{t,p} \in I, t \in T, p \in P, I = I^+ \cup I^-\}$ is the edge set.

More formally, for the Graph $\mathscr{G}(\mathscr{V}, \mathscr{E})$, the input features of TCRs or peptides are $\{x_v, \forall v \in \mathscr{V}\}$, and the edges $\{\forall e \in \mathscr{E}\}$ represent the relationships between TCRs and peptides, whether positive or negative. The TCR or peptide embedding $z_v$ generation algorithm is described as follows:

$$h_v^0 = x_v, \forall v \in \mathscr{V} \tag{1}$$

$$h_{\mathscr{N}(v)}^k = AGGREGATE\left(\{h_u^{k-1}, \forall u \in \mathscr{N}(v)\}\right), \forall k \in \{1, \ldots, K\} \tag{2}$$

$$h_v^k = \sigma\left(CONCAT\left(W_k h_{\mathscr{N}(v)}^k, B_k h_v^{k-1}\right)\right), \forall k \in \{1, \ldots, K\} \tag{3}$$

$$z_v = h_v^K, \forall v \in \mathscr{V} \tag{4}$$

where $h_v^k$ is the representation of node $v$ at k-th order. $h_v^0$ is initialized by the input features of TCRs or peptides. $\mathscr{N}(v)$ denotes the neighbors of node $v$. $z_v$ is final representations of node $v$. $W_k$ and $B_k$ are trainable weight matrices, the depth K is 3 by default and the non-linearity $\sigma$ is Leaky ReLU. The differentiable aggregator functions $AGGREGATE$ is the average of neighbor's previous layer embeddings. $h_{\mathscr{N}(v)}^k$ is trivially computed by $h_{\mathscr{N}(v)}^k = \sum_{u \in \mathscr{N}(v)} \frac{h_u^{k-1}}{|\mathscr{N}(v)|}$ with $|\cdot|$ donating the number of the node $v$ neighbors. To generate the representation of node $v$, it first aggregates the representations of $\mathscr{N}(v)$ and then updates the representation of $v$ by concatenating its representation at (k-1)-th order and the aggregated representations.

## MLP classifier
A linear layer transforms one vector into another vector, which refers to a single-layer neural network without hidden layers. The MLP is a stack of linear layers with hidden layers. For classification tasks, HeteroTCR uses an MLP with two hidden layers (Fig. 1). The two numerical vectors (2048 dimension) from the Heterogeneous GNN module are concatenated into a single layer with 512 neurons activated by ReLU, followed by a dense layer with 256 neurons also activated by ReLU, and the last layer with a single neuron with sigmoid activation. Mathematically, the output is a continuous variable between 0 and 1, representing the predicted binding strength between TCR and peptide. HeteroTCR was trained for 1000 epochs with 512 batch sizes using the Adam optimizer with a learning rate of 0.0001 and a weight decay of 0 by default. The loss function is binary cross-entropy.

## Levenshtein similarity ratio
Here, the Levenshtein ratio was used as a measure of the similarity between TCR sequences. The Levenshtein ratio is based on the *ldist*, which is not Levenshtein distance, but the sum of the costs. Given two strings, the *ldist* describes the count of modifications needed to transform one word into another. The possible changes are deletion, insertion, and replacement. The count of deletion and insertion is 1, while the count of replacement is 2. The Levenshtein ratio is given by the formula

$$Ratio_{Lev} = \frac{sum(|u|, |v|) - ldist(u, v)}{sum(|u|, |v|)} \tag{5}$$

where $u$ and $v$ represent two TCR sequences, and $|\cdot|$ defines their length.

We defined the Levenshtein similarity ratio of a TCR as the average of the Levenshtein ratio from the TCR to each TCR in the training set, and then removed the data exceeding the specified ratio cut-offs. In Fig. 6a, we calculated the average of each TCR's Levenshtein ratio in the training set to all of the training TCRs and thus reduced TCR similarity within the training set. In Fig. 6b we calculated the average of each TCR's Levenshtein ratio in the testing set to all of the training TCRs and thus reduced the TCR similarity between the testing set and the training set.

## Data curation of the 10× Genomics platform and calculation of binding fractions
The data utilized in our study was generated from the 10× Genomics Chromium Single Cell Immune Profiling platform (https://www.10xgenomics.com/resources/datasets). The raw data is accessible for download at https://zenodo.org/records/6952657. Closely examining four single-cell datasets, we analyzed profiles of CD8$^+$ T cells specific to a highly multiplexed panel consisting of 44 different pMHC multimers, alongside 6 negative control pMHC multimers, sourced from four healthy donors. The binding specificity between T cells and each pMHC complex was quantified using the UMI counts as a binding indicator.

In general, a T cell typically expresses only one pair of functional TCRs, and thus, we selectively retain clones of T cells expressing a single pair of TCR$\alpha$ and TCR$\beta$ chains. We focused only on expanded clones, as the UMI counts of each cell were inherently noisy due to dropouts and high variances in single-cell experiments. Consequently, we opted to utilize the binding fraction of each T cell clone with the same TCR as a measure of antigen affinity to a TCR. The binding fraction of a clone is determined by the following formula:

$$Binding\ Fraction = \frac{m}{n} \tag{6}$$

where $m$ represents the number of T cells within the clone exhibiting a higher UMI count for a given antigen compared to the maximum UMI count among the 6 negative control antigens, and $n$ denotes the clone size, i.e., the number of T cells with identical TCRs. It is noteworthy that instances where $m$ equals 0 are only considered in the following two scenarios: when the UMI count of T cells within the clone for a given antigen is less than the maximum UMI count among the 6 negative control antigens, or when the UMI count of T cells within the clone for a given antigen equals the maximum UMI count among the 6 negative control antigens, and both are non-zero. Specifically, we do not consider cases where $m$ equals 0 when the UMI count of T cells within the clone for a given antigen is 0, and the maximum UMI count among the 6 negative control antigens is also 0. This omission is due to the sparse nature of the original data, characterized by numerous zero values, leading to excessive noise and inclusion of many meaningless data points. Finally, we calculated the Spearman correlation coefficient between predicted binding probabilities and binding fractions.

## Machine learning platform and hardware
The pre-trained CNN module was implemented with Keras 2.6.0 (https://keras.io) using the Tensorflow backend and Python 3.7.0. The Heterogeneous GNN module and the MLP classifier were implemented with PyTorch 1.9.1 (https://pytorch.org) using Python 3.7.11. PyG (PyTorch Geometric)[37], a library built upon PyTorch to easily write and train GNNs, was employed for modeling and processing graph-structured data. The metric evaluation was implemented with TorchMetrics package[40]. All deep learning models were trained on a single NVIDIA A40 graphics card.

## Reporting summary
Further information on research design is available in the Nature Portfolio Reporting Summary linked to this article.

## Data availability

All additional data that support the conclusions from this manuscript are available in the Supplementary Information. Source data and codes for the main figures, as well as all data utilized for HeteroTCR training and evaluation, are shared on the github repository (https://github.com/yuzilan/HeteroTCR) and its Zenodo (https://doi.org/10.5281/zenodo.11120879). The dataset used for training and testing were collected from IEDB, VDJdb (https://vdjdb.cdr3.net) and McPAS-TCR (http://friedmanlab.weizmann.ac.il/McPAS-TCR/). The detailed information of the 10x Genomics cohort is available at https://www.10xgenomics.com/resources/datasets. The raw data for single-cell datasets is available at https://zenodo.org/records/6952657.

## Code availability

HeteroTCR is available on github (https://github.com/yuzilan/HeteroTCR) and its Zenodo (https://doi.org/10.5281/zenodo.11120879), together with a usage documentation and several example testing datasets.

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

## Acknowledgements

This work was funded by grants from the Tsinghua University Independent Research Project (ID: 52302102423), the Tsinghua University Spring Breeze Fund, and the Beijing Institute of Technology's Proof of Concept Project for Tumor Neoantigen Personalized TCR-T Therapy.

## Author contributions

Zilan Yu, Mengnan Jiang, and Xun Lan designed the framework of this study. Zilan Yu devised the model architecture and implemented the code. Mengnan Jiang contributed biological insights, created figures, and acquired and preprocessed data for analysis. Zilan Yu wrote the paper with the help of other authors. Xun Lan supervised the study. All authors read and approved the final paper.

## Competing interests

The authors declare no competing interests.
