## [Peer Review File · Communications Biology]

Reviewers' comments:

Reviewer #1 (Remarks to the Author):

General comments

The authors proposed HeteroGNN for peptide-TCR binding prediction. The model consisted of a convolutional neural network followed by a heterogenous graph neural network. The latter learned the interactions between TCR and peptides, as well as the similarity between sequences of one type related to the same sequence of the other type. The graph neural network was shown to play an important role in the good and generalized prediction of TCR-peptide interactions. HeteroGNN prediction probability was also aligned with the pMHC binding affinity, which confirmed the potential application of the tool in immunotherapy research.

The manuscript was well-written and easy to follow. Every section was organized with a good amount of information provided. However, some points could be addressed for an improved version of the manuscript.

Comment 1:

In Figure 3, HeteroGNN showed good performance on the TCR-based test set, which highlighted the model's ability to learn the relationship between TCRs. The model also performed well on the antigen-based test set, which showed the efficiency of the model in learning the peptide-peptide interconnections. Then I am wondering about the reason why the model's performance significantly dropped on the strict-based test set. Should one assume that the information captured from intra-type and inter-type relations was enough for predicting interactions of TCR and peptide unseen from the training set? What other factors could affect the model's performance in such cases?

Comment 2:

In part 2.5, the authors showed nice examples of how the TCR-peptide pairs cluster from the information learned by the graph neural network. It also clearly showed an improved performance of the prediction across different groups of peptides. However, the impact of the GNN module was unclear considering the TCR aspect.

Indeed, the information from Figure 5d was in line with the model's performance in the TCR-based test set (Figure 3). The model worked well with TCR unseen from the training set and with different degrees of similarity between TCR sequences. However, it was unclear to me how to link the message shown in Figures 5a, b, and c with the one in Figure 5d. Maybe the author can clarify this point.

Comment 3:

The GitHub repository of the tool should be indicated specifically instead of the general Github address. I tried to look up the repository of HeteroGNN but I couldn't.

Comment 4:

Figures should be provided of a higher quality. The texts, numbers, and symbols in the figures should be increased in size. The resolution of the figures was limited, so the texts were visually broken when zoomed in.

The caption of the figures should be enlarged as well.

Comment 5:

In Figure 1, I can understand the graph in green shade and in blue shade, which corresponded to the node information aggregated from the first, second, and third-order neighborhoods of the circle and the triangle nodes, respectively. But this would challenge the readers since the information was not clearly explained in the text, and the "k=1", "k=2", "k=3", and the gray circles were not visible enough.

Comment 6:

"For the TCR-based data sets, the TCRs present alone in the training dataset were removed from the testing dataset. For the antigen-based data sets, the peptides present alone in the training dataset were removed from the testing dataset. For the strict-based data sets, both TCRs and peptides present alone in the training dataset were removed from the testing dataset."

"...present alone... were removed" might not be the most appropriate way to convey the author's intention. Please rewrite to improve the readability of the text.

Reviewer #2 (Remarks to the Author):

This paper introduces an approach for prediction of peptide-TCR recognition using heterogeneous Graph neural network. The authors claim that the proposed method can capture interactions between peptides and TCRs as well as similarities within TCRs or peptides, and thus improve the prediction accuracy. The paper is well-written and organized. The results are discussed in detail and the comparison with existing methods is comprehensive. Generally, the work has originality and has potential prospects of peptide-TCR exploration. I believe it would be helpful for the studies on the adaptive immune response.

Specific comments for revision:

1. For section 2.3, TCR-based data splitting method is stricter than paired-based, but the performance on TCR-based test data is better than paired-based. I expect the authors to provide some explanation on this point.
2. The authors compared the model-extracted representations of peptide-TCR pairs with or without GNN module. I suggest the authors to provide more numerical comparison and evidence on this part, especially for the part that GNN module can learn the similarities within TCRs or peptides. For example, for a peptide (TCR), are the distances in the GNN embedded space smaller than distances in the CNN embedded space among TCRs (peptide) corresponding with this peptide (TCR)?
3. What is the model's sensitivity to the parameter K? Will the model's performance increase when K is larger?
4. For figure 2, there are only five data points for each boxplot. It may be more straightforward for the comparison to plot all data points.
5. For section 2.4, for the comparison with TITAN, it would be helpful to report the size of test data.
6. Table 1, 'Unique-epitope data' -> 'Unseen-epitope data'

Reviewer #3 (Remarks to the Author):

Brief Summary

Yu, Jiang, and Lan endeavor to leverage a class of machine learning approaches graph neural networks (GNN) to the predictive task of TCR-pMHC recognition. This a notoriously difficult prediction task given enormous diversity of peptides & TCRs, as well as fact that TCRs may recognize multiple peptide MHC ligands. The authors appear to demonstrate an improvement in performance over existing tools, but the manuscript could be strengthened by providing additional intuitive explanation of GNN and with more detailed description of the experimental validation (section 2.6). Code and data, with sufficient documentation on computing environment and ML platform to facilitate reproduction, should be made fully available prior to publication. Code was not made available at the time of this review.

Overall impression

A. Yu et al. do a nice job of categorizing the wide variety of existing models for the task predicting molecular TCR-pMHC. The introductory section -- Lines 43-63 are very clear and well written, distinguish categorical and generic model. This well motivates the work.

B. The authors carefully considering the type of data splitting and how presence identical TCR or peptides in the training and testing set influence model performance. References to other “state-of-the-art” methods is appropriate and helpful, however, few methodological details about how other benchmark methods were implemented is provided. While providing mathematical details of method, little practical methodological guidance is provided about machine learning platform or hardware used to implement the GNN.

C. Many readers of Communications Biology will certainly be unfamiliar with GNNs, and additional motivation and description may be helpful. For instance, in what other areas have GNNs been leveraged for prediction? Have they been used previously in biology or protein interaction studies? Do the authors cite a good background reference for interested readers? Can author provide some intuition to the lay person about how the heterogenous GNN module incorporates a new unknown sequences or peptides in the prediction task? It is somewhat clear how the graph could be formed from training data but how are new unlabeled data incorporated?

D. The study mentions “ablation studies” and “visualization” substantiating the essential role of the GNN in finding feature underlying binding process. In this dimension the manuscript could be strengthened. Frist, the visualization based on tSNE are hard to interpret and it’s not clear if the claims could be falsified based on their inspection alone. Additionally, the authors use “ablation studies” to mean removal of components of the AI system to assess changes in performance. This is rational, but this word choice could also be confusing to biologists who may expect “ablation” to mean changing predicted residues of a peptide or TCR to test binding predictions. While this experimental work may be out of scope for a computational effort, could the authors use their tool to generate specific falsifiable predictions about which residues could be changed to abrogate TCR-pMHC reactivity? Alternatively, can the “model extracted representations” be visualized in any way to enable human interpretable understanding CDR3 feature essential for peptide recognition?

E. The authors assert that their method outperforms existing models. In addition to comparison made, can the authors contextualize the gain in performance? Given that this is a very challenging prediction task with a limited amount of training data, can the authors clarify whether the performance achieved (AUC = .6535, strict data splitting, figure 3) might allow in terms of any practical application? Does the present work provide some indication on the amount or quality of training data that would be necessary for the model to achieve a higher level of accuracy? For instance, what if half as much training data had been used? What sort of data would be most valuable for increasing generalizable prediction? Does the heterodimeric nature of the TCR receptor place expected limits on any method based on the CDR3 alone?

F. While the tool may represent a computational advance, the application section 2.6 is very cursory and may not justify the full claims of the tools utility made in the discussion.

Specific Comments

1. The author suggest that their code is made publicly available. The link in the manuscript is a generic link to GitHub and does not provide access for reviewers to evaluate. Code for this work should be made available to reviewers in advance and must be made available prior to publication.

2. Line 70 Please provide additional explanation of how the GNN extracts “within-type” information about “TCR-TCR” or “peptide-peptide”. Please provide the reader with intuition on how the training data (typically a TCR to a pMHC ligand assignment) also provides information about “peptide-peptide” similarity? Clarify line 123. What is meant by “This implies that TCRs (peptides) associated with the same peptide (TCR) may similarity to some extent, making it possible to capture within-type (TCR-TCR or peptide-peptide) similarity”. I found this sentence confusing. Does this require training data where two peptides are recognized by the same CDR3b sequence? If so how common is this in the training

data?

3. Line 150: Why is 0.5 an appropriate threshold for predicting interaction?

4. Line 156: word "ratating" perhaps "rotating"

5. Fig 2: consider removing gray line (trend in model performance, seems irrelevant).

6. Fig 2: What are the repeated measures in each boxplot, was 5-fold data partition of the training data used to create a separate model that was then applied to the VDJdb testing data?

7. Line 179: what does "alone" in "TCRs present alone in the training set" mean. Does this just mean that exact TCR (CDR3b) sequences present in the training data were removed from the testing dataset? Please clarify the data-splitting rule description Line 178-182. Description in methods is clearer.

8. Fig 3: An interesting feature of Figure 3 is the low variance in AUC for HeteroTCR vs other methods. What might explain this?

9. Line 247: "Further evaluation took place independently on VDJdb with a confidence score greater than or equal to 0" – Since 0 is the lowest confidence score, doesn't this just imply all of VDJdb was used regardless of confidence score?

10. Line 259 -How is model representation embedded with tSNE, could more details be provided. What is the dimension of data being embedded? Figure 5 shows with and without GNN module, but it is very hard to understand what exactly the "model-extracted representations of peptide-TCR pairs" are. Consider a supplementary diagrammatic figure that could provide more detail than Fig 1 on the dimensions of vectorized representations with and without GNN?

11. It is hard to see "distinct boundary" in Fig 5a (right). Maybe a close up figure of the relevant points or another term is more appropriate.

12. Line 290: the word "decrease" is likely misspelled

13. Line 286, why use Levenshtein ratio instead of distance? Figure 5d is confusing, one would expect more data at further Levenshtein distance but the red line suggests that there is less data at further distance? Is this correct? Also, it would be much more interpretable to a biologist to explain that the model can generalize even to TCRs X-X "edit distance" away from the nearest training set TCR? The ratio is explained in methods, but can more information be provided for interpretation.

14. Consider an alternative term for "Real-life validation", perhaps something like "GNN binding predictions correlate with experimentally derived TCR pMHC affinity"

15. Overall, section 2.6 is extremely brief and somewhat confusing

16. Correlation between GNN predicted score and binding affinity (which is probably noisily measured in this context) is low but positive. Is the spearman correlation based on the bins or all of the data. Why not show the data as a scatter plot? What do the error bars in Fig 6 indicate?

17. Line 331: The authors should be careful not to overstate the utility of existing tool and extent of validation. No effort was made to identify neo-epitopes based on HeteroTCR predictions.

Point-by-point Response to Reviewer's Comments

We thank the reviewers for their valuable feedback on our manuscript. Their constructive suggestions have significantly enhanced the quality of our study. In the sections below, we present a detailed point-by-point response that includes summaries of changes made in response to these insightful comments. The original reviewers' comments are colored in black, and our responses are provided in blue. The changes in the manuscript are highlighted in yellow. We sincerely hope that these revisions align with the expectations of the anonymous reviewers.

Reviewer #1:

General comments

The authors proposed HeteroGNN for peptide-TCR binding prediction. The model consisted of a convolutional neural network followed by a heterogenous graph neural network. The latter learned the interactions between TCR and peptides, as well as the similarity between sequences of one type related to the same sequence of the other type. The graph neural network was shown to play an important role in the good and generalized prediction of TCR-peptide interactions. HeteroGNN prediction probability was also aligned with the pMHC binding affinity, which confirmed the potential application of the tool in immunotherapy research.

The manuscript was well-written and easy to follow. Every section was organized with a good amount of information provided. However, some points could be addressed for an improved version of the manuscript.

Response: Thank you very much for your kind words and positive feedback on the manuscript.

Comment 1:

In Figure 3, HeteroGNN showed good performance on the TCR-based test set, which highlighted the model's ability to learn the relationship between TCRs. The model also performed well on the antigen-based test set, which showed the efficiency of the model in learning the peptide-peptide interconnections. Then I am wondering about the reason why the model's performance significantly dropped on the strict-based test set. Should one assume that the information captured from intra-type and inter-type relations was enough for predicting interactions of TCR and peptide unseen from the training set? What other factors could affect the model's performance in such cases?

Response: Thank you for your comment. We believe that the significant drop in the model's performance on the strict-based test set can be attributed to the limited number of antigens in the databases, which only represent a small portion of the possible antigen space. Achieving high performance on unseen antigens is particularly difficult for models trained on a small number of antigens, which we consider as one of the most challenging issues in the field at present. To address this issue and expand the training data, we are actively developing yeast mating-based methods (Wang et al., 2022, Cell Discovery) and yeast adhesion-based screening techniques. These approaches aim to identify interacting peptide-TCR pairs in a high-throughput manner. Another crucial factor is the complete absence of TCRs and antigens from the training data in the strict-based test set. When applying the model to the test pairs, there were no direct "hints" on the edges of the new pairs in the graph model from previously seen antigens or TCRs.

Comment 2:

In part 2.5, the authors showed nice examples of how the TCR-peptide pairs cluster from the information learned by the graph neural network. It also clearly showed an improved performance of the prediction across different groups of peptides. However, the impact of the GNN module was unclear considering the TCR aspect. Indeed, the information from Figure 5d was in line with the model's performance in the TCR-based test set (Figure 3). The model worked well with TCR unseen from the training set and with different degrees of similarity between TCR sequences. However, it was unclear to me how to link the message shown in Figures 5a, b, and c with the one in Figure 5d. Maybe the author can clarify this point.

Response: We agree with the reviewer that the connection between Figure 5a, b, c and Figure 5d is somewhat weak. Specifically, Figures 5a, b, c illustrate the influence of the GNN module on the model's performance, while Figure 5d aims to demonstrate the robustness of HeteroTCR in predicting TCRs with amino acid sequences dissimilar from those in the training data. In response to this feedback, we have restructured the manuscript by separating Figure 5d from the other panels, designating it as the new Figure 6.

Comment 3:

The GitHub repository of the tool should be indicated specifically instead of the general Github address. I tried to look up the repository of HeteroGNN but I couldn't.

Response: We apologize for the mistake. The GitHub repository for HeteroTCR was initially set to private. We have now made the code public. You can find the repository at <https://github.com/yuzilan/HeteroTCR>. Please feel free to check it out.

Comment 4:

Figures should be provided of a higher quality. The texts, numbers, and symbols in the figures should be increased in size. The resolution of the figures was limited, so the texts were visually broken when zoomed in.

The caption of the figures should be enlarged as well.

Response: Thank you for pointing this out. The conversion to PDF from word document resulted in a loss of image quality. In the revised manuscript, we have followed the Submission guidelines and uploaded each image separately to ensure clarity. We also enlarged the captions of the figures.

Comment 5:

In Figure 1, I can understand the graph in green shade and in blue shade, which corresponded to the node information aggregated from the first, second, and third-order neighborhoods of the circle and the triangle nodes, respectively. But this would challenge the readers since the information was not clearly explained in the text, and the "k=1", "k=2", "k=3", and the gray circles were not visible enough.

Response: Thank you for pointing this out. To address this concern, we have revised the text to provide a more explicit explanation of the depicted elements and enhanced the visibility of the relevant annotations in the figure.

Revised manuscript (Page 3, Line 94-96):

Fig. 1. Architecture of HeteroTCR. **Left,** Overview of pre-trained CNN module architecture (Methods). **Middle,** Visual illustration of Heterogeneous GNN (Methods). A circle node represents a TCR, while a triangle node represents a peptide. The edge of the solid line represents an interaction between a TCR and a peptide, while the dotted line represents no interaction, and no connection represents unknown. It is important to note that when initializing the graph, the nature of the relationship between peptides and TCRs is unknown, and we can only rely on the input training pairs to identify the existence of a relationship. We get the weight matrix by iterating 1000 epochs of training to learn and adjust the nature of the relationship between the given TCR and peptide, whether it is a solid line or a dotted line. We use a set of aggregator functions to learn aggregate feature information from K^{th} -order neighborhoods of a node, with a default K value of 3. **As shown in the graphs with green and blue shading, “ $k=1$ ”, “ $k=2$ ”, “ $k=3$ ”, represent the node information aggregated from the first, second, and third-order neighborhoods of the circle and triangle nodes.** In other word, a given peptide/TCR can learn information from its 1st-order binding TCRs/peptides, while the 1st-order TCR/peptide can learn information from its 1st-order binding peptides/TCRs which are 2nd-order neighborhoods of given peptide/TCR. Therefore, a given peptide/TCR can extract between-type (peptide-TCR) interaction information from the 1st- and 3rd-order TCRs/peptides, and obtain within-type (TCR-TCR or peptide-peptide) similarity information from the 2nd-order peptides/TCRs. This is because TCRs/peptides with shared features tends to interact with the same peptide/TCR. This process continues until order K is reached. **Right,** Overview of MLP in HeteroTCR (Methods). Overall, HeteroTCR is composed of three main components: a pre-trained CNN, a Heterogeneous GNN, and an MLP classifier. The numeric embeddings extracted from the pre-trained CNN module are utilized as inputs for the Heterogeneous GNN module, which enables the extraction of information from the sequences. Specifically, the Heterogeneous GNN initially creates a global graph based on the entire dataset, and then trains on each pair of peptide-TCR inputs.

Comment 6:

“For the TCR-based data sets, the TCRs present alone in the training dataset were removed from the testing dataset. For the antigen-based data sets, the peptides present alone in the training dataset were removed from the testing dataset. For the strict-based data sets, both TCRs and peptides present alone in the training dataset were removed from the testing dataset.”

“...present alone... were removed” might not be the most appropriate way to convey the author’s intention. Please rewrite to improve the readability of the text.

Response: Thank you for pointing this out. We’ve revised this paragraph as follows for clarity:

"For TCR-based data sets, peptide-TCR pairs involving TCRs present in the training dataset are excluded from the testing dataset. Similarly, for antigen-based data sets, peptide-TCR pairs involving peptides appearing in the training dataset are excluded from the testing dataset. In the case

of strict-based data sets, peptide-TCR pairs involving either TCRs or peptides present in the training dataset are excluded from the testing dataset."

Revised manuscript (Page 6, Line 200-204):

To further demonstrate the generalizability of HeteroTCR, we evaluated models based on four different types of data splitting methods (Methods), namely pair-based, TCR-based, antigen-based, and strict-based data sets. For the pair-based data sets, the peptide-TCR pairs present in the training dataset were removed from the testing dataset. For TCR-based data sets, peptide-TCR pairs involving TCRs present in the training dataset are excluded from the testing dataset. Similarly, for antigen-based data sets, peptide-TCR pairs involving peptides appearing in the training dataset are excluded from the testing dataset. In the case of strict-based data sets, peptide-TCR pairs involving either TCRs or peptides present in the training dataset are excluded from the testing dataset.

Reviewer #2:

This paper introduces an approach for prediction of peptide-TCR recognition using heterogeneous Graph neural network. The authors claim that the proposed method can capture interactions between peptides and TCRs as well as similarities within TCRs or peptides, and thus improve the prediction accuracy. The paper is well-written and organized. The results are discussed in detail and the comparison with existing methods is comprehensive. Generally, the work has originality and has potential prospects of peptide-TCR exploration. I believe it would be helpful for the studies on the adaptive immune response.

Response: Thank you very much for your kind words and positive feedback on the manuscript.

Specific comments for revision:

Comment 1:

For section 2.3, TCR-based data splitting method is stricter than paired-based, but the performance on TCR-based test data is better than paired-based. I expect the authors to provide some explanation on this point.

Response: We sincerely appreciate your thought-provoking feedback. We are not quite sure about the exact reasons, however, here is our hypothesis. In the available databases, a single peptide usually interacts with multiple TCRs, while a TCR typically interacts with one or two peptides. The majority of the peptide-TCR pairs were derived from experimental screening. TCRs interact with multiple peptides might be overrepresented TCR sequences that were false positives in the original screening. Excluding duplicate TCRs in the testing set might help mitigate potential noises and therefore increase the robustness and the generalizability of the model. To test this hypothesis, we removed all TCRs interact with 2 or more antigens in both the training and the testing data. Our results showed that by removing of all duplicated TCRs our model achieved an AUC of 0.7359 (std=0.0039), better than that under TCR-based splitting (AUC=0.7154, paired t-test P-value between the AUC of TCR-based splitting dataset and that of dataset with duplicated TCRs removed is < 0.001).

Comment 2:

The authors compared the model-extracted representations of peptide-TCR pairs with or without GNN module. I suggest the authors to provide more numerical comparison and evidence on this part, especially for the part that GNN module can learn the similarities within TCRs or peptides. For example, for a peptide (TCR), are the distances in the GNN embedded space smaller than

distances in the CNN embedded space among TCRs (peptide) corresponding with this peptide (TCR)?

Response: Thank you for your valuable suggestions on our manuscript. In the revised manuscript, we introduce additional numerical metrics that measure the distances of the colored points in the GNN embedded space compared to those in the CNN embedded space, as illustrated in Fig. 5c (below). Fig. 5d, previously labeled as Fig. 5c in the initial version of the manuscript, depicts the AUC of models with and without the heterogeneous GNN module. This figure underscores that positive and negative samples are more easily separable in the GNN embedded space than in the CNN embedded space.

Revised manuscript (Page 10, Line 324-332):

Fig. 5. Superiority and interpretability of the design of HeteroTCR. a-b, The t-SNE plot for the peptide-TCR pairs of the given peptide with and without a Heterogeneous GNN module. Each point represents a pair of peptide-TCR and is colored purple or orange, where purple represents no interaction and orange represents an interaction. Without Heterogeneous GNN module, the colored dots are scattered, and no clear boundary can be observed between purple and orange dots. In contrast, with Heterogeneous GNN, the colored dots are more clearly clustered and classified. c, The degree of aggregation of the colored points (peptides with ≥ 15 cognate TCRs) is assessed by computing the center of the point set and the average distance of each point to the center. d, The per-peptide (with ≥ 15 cognate TCRs) AUC for the models with and without Heterogeneous GNN module, which are trained five times on IEDB and tested on VDJDdb with confidence score greater than or equal to 1, based on the pair-based setting.

An inherent challenge in generic peptide-TCR models is their potential to memorize TCR motifs regardless of the peptide partner, leading to an inability to capture the true underlying molecular forces governing the binding process and merely managing to memorize spurious motifs present within the TCR training data [23]. To assess whether HeteroTCR captures essential binding features or memorizes spurious TCR motifs, we visualized model-extracted representations of peptide-TCR pairs in a 2-dimensional space using t-distributed stochastic neighbor embedding (t-SNE) [36] (Fig. 5a-b). Due to the large number of peptide-TCR pairs in VDJDdb with a confidence score greater than or equal to 0, we selected representative visuals from VDJDdb with a confidence score greater than or equal to 1. Notably, using numeric representations extracted by the full model, cognate TCRs for a specific antigen clustered more tightly compared to a model without the Heterogeneous GNN module. Additionally, with the Heterogeneous GNN module, we observed a

better separation between the orange dots (representing positive samples, indicating peptide-TCR interactions) and the purple dots (indicating negative samples, denoting no peptide-TCR interactions).

To measure the degree of aggregation of positive and negative samples for the same antigen, we calculated the center of the point set and the average distance of each point to the center. We found that the degree of aggregation of the samples with the Heterogeneous GNN module, is significantly smaller than that without the Heterogeneous GNN module, indicating that cognate TCRs for a specific antigen cluster more tightly (paired t-test P-value between the models with and without Heterogeneous GNN module is 3.9×10^{-10}) (additional details in Supplementary Table 8). Fig. 5d shows the AUC in per-peptide classification of positive and negative samples (≥ 15 cognate TCRs), demonstrating the Heterogeneous GNN module's reasonable performance (paired t-test P-value between the models with and without Heterogeneous GNN module is 1.2×10^{-7}) (additional details in Supplementary Table 9). This observation reveals that the information on between-type (peptide-TCR) interactions and within-type (TCR-TCR or peptide-peptide) similarity learned by HeteroTCR prevents reliance on TCR motif patterns for interaction prediction.

Comment 3:

What is the model's sensitivity to the parameter K? Will the model's performance increase when K is larger?

Response: Thank you for raising the question. In the revised manuscript, we conducted experiments to assess the model's performance in relation to parameter K, as detailed in Section 2.5. Our findings indicate a non-linear relationship between K and model performance, characterized by an initial improvement followed by a subsequent decline. Notably, the model achieves the highest AUC at K=4. However, considering the associated increase in training depth and computational cost at higher K values, we recommend adopting K=3 as it provides a balanced trade-off between time complexity and performance.

Revised manuscript (Page 9, Line 293-299):

Furthermore, we investigated the relationship between model performance and the parameter K (Table 3, additional data information available in Supplementary Table 7). We considered K values ranging from 1 to 6. Our findings indicate a non-linear relationship between K and model performance, characterized by an initial improvement followed by a subsequent decline. Notably, the model achieves the highest AUC at K=4. However, considering the associated increase in training depth and computational cost at higher K values, we recommend adopting K=3 as it provides a balanced trade-off between time complexity and performance.

Table 3. Model's sensitivity to the parameter K

K	AUC (std)
1	0.6686 (0.0035)
2	0.6895 (0.0030)
3	0.7027 (0.0039)
4	0.7157 (0.0011)
5	0.7084 (0.0052)
6	0.7047 (0.0028)

Comment 4:

For figure 2, there are only five data points for each boxplot. It may be more straightforward for the comparison to plot all data points.

Response: Thank you for your valuable suggestion. In the revised manuscript, we superimposed all data points on the boxplot.

Revised manuscript (Page 6, Line 190-194):

Fig. 2. Performance comparison between HeteroTCR and published methods based on pair-based data sets with 5-fold cross-validation. All boxplots: The center line of each boxplot marks the sample median, and the box extends from the lower to upper quartile. Each colored point represents one data point. Our results show that HeteroTCR outperforms tested methods, as indicated by a paired t-test P-value of < 0.00001 for comparisons with any other model.

Comment 5:

For section 2.4, for the comparison with TITAN, it would be helpful to report the size of test data.

Response: Thank you for your valuable suggestion. In the revised manuscript, we added information about the size of the test data for both ImRex and TITAN.

Revised manuscript (Page 8, Line 252-254):

Table 1. Comparison of AUC between HeteroTCR and ImRex

Settings	ImRex	HeteroTCR	Number of samples in training data (positive/negative)	Number of samples in testing data (positive/negative)
Shared-epitope data	0.59	0.69	6,702/6,702	4,101/4,101
Unique-epitope data	0.54	0.61	6,702/6,702	736/736

Note: Best performance marked in bold. Data of ImRex is referenced from [23]. We generated the same number of negative samples for each dataset using the a random shuffling approach.

Revised manuscript (Page 8, Line 266-273):

Table 2. Comparison of AUC between HeteroTCR and TITAN

Models	TCR split		Strict split	
	Training data (positive/negative)	Testing data (positive/negative)	Training data (positive/negative)	Testing data (positive/negative)

	9,539/9,539	1,060/1,060	9,171/9,171	1,008/1,008
TITAN K-NN	0.79 ± 0.01		0.54 ± 0.03	
TITAN AA CDR3	0.75 ± 0.02		0.60 ± 0.04	
TITAN AA full	0.76 ± 0.007		0.59 ± 0.04	
TITAN SMI CDR3	0.73 ± 0.007		0.60 ± 0.06	
TITAN SMI full	0.75 ± 0.006		0.59 ± 0.06	
TITAN Pretrained	0.81 ± 0.01		0.56 ± 0.04	
TITAN Pretrained aug.	0.80 ± 0.01		0.59 ± 0.03	
TITAN Pretrained semifrozen	0.80 ± 0.01		0.58 ± 0.06	
TITAN Pretrained semifrozen aug.	0.82 ± 0.01		0.62 ± 0.06	
HeteroTCR	0.83 ± 0.01		0.67 ± 0.09	

Note: Data of TITAN is referenced from [7]. Mean and standard deviation of each model configuration on TCR split and strict split. Best performance marked in bold. *K-NN* refers to the baseline model of TITAN. *AA* means sequences are encoded by amino acid while *SMI* refers to peptides that are encoded by SMILES. *CDR3* denotes only CDR3 sequences of TCRs are input of the model while *full* denotes full sequences of TCRs are feed to the model. All TITAN Pretrained models adopt SMILES encodings of peptides and full sequence input for TCRs. *Pretrained* means the model is pre-trained on BindingDB, *aug* means the model is pre-trained on BindingDB with data augmentation, and *semifrozen* means the weights in the epitope channel are fixed during fine-tuning.

Comment 6:

Table 1, ‘Unique-epitope data’ -> ‘Unseen-epitope data’

Response: Thank you for pointing this out. We have revised this terminology as you suggested. We added a note indicating that the ImRex paper used the nomenclature “unique-epitope” which is equivalent to “unseen epitope” used here.

Revised manuscript (Page 7, Line 249-250):

In ImRex (Table 1), the VDJdb dataset was used for training with 5-fold cross-validation, and the McPAS-TCR dataset served as an independent testing set, which was split into two subsets: one containing peptides already present in the training set (shared-epitope data), and the other containing peptides not seen during ImRex training (unseen-epitope data, which was referred as the “unique-epitope data” by the authors of ImRex [23]). Negative interaction pairs were generated using a shuffling approach for each subset. We observed that HeteroTCR consistently outperformed ImRex on both independent testing sets.

Reviewer #3:

Brief Summary

Yu, Jiang, and Lan endeavor to leverage a class of machine learning approaches graph neural networks (GNN) to the predictive task of TCR-pMHC recognition. This a notoriously difficult prediction task given enormous diversity of peptides & TCRs, as well as fact that TCRs may recognize multiple peptide MHC ligands. The authors appear to demonstrate an improvement in performance over existing tools, but the manuscript could be strengthened by providing additional intuitive explanation of GNN and with more detailed description of the experimental validation (section 2.6). Code and data, with sufficient documentation on computing environment and ML platform to facilitate reproduction, should be made fully available prior to publication.

Code was not made available at the time of this review.

Response: Thank you for your thorough review and valuable insights on our manuscript.

Overall impression

A. Yu et al. do a nice job of categorizing the wide variety of existing models for the task predicting molecular TCR-pMHC. The introductory section -- Lines 43-63 are very clear and well written, distinguish categorical and generic model. This well motivates the work.

Response: Thank you for your kind words and positive feedback.

B. The authors carefully considering the type of data splitting and how presence identical TCR or peptides in the training and testing set influence model performance. References to other “state-of-the-art” methods is appropriate and helpful, however, few methodological details about how other benchmark methods were implemented is provided. While providing mathematical details of method, little practical methodological guidance is provided about machine learning platform or hardware used to implement the GNN.

Response: Thank you very much for pointing this out. We apologize for the lack of details and practical guidance in the submitted manuscript. In the revised manuscript, we provide details on the implementation of benchmark methods, as well as the machine learning platform and hardware used for implementing the GNN, hoping to enhance the clarity and reproducibility of the results.

Revised manuscript (Page 5, Line 163-181):

We assessed the generalizability of HeteroTCR and other published methods on pair-based data sets (Methods). Specifically, the five models under comparison include NetTCR-1.0, NetTCR-2.0, ERGO LSTM, ERGO AE, and DLpTCR. NetTCR-1.0 [20] is based on simple 1D convolutional neural network (CNN), integrating peptide and CDR3 β sequence information encoded with BLOSUM50 matrix for the prediction of peptide-TCR specificity. NetTCR-2.0 [5] modifies the network structure on the basis of NetTCR-1.0 and uses paired CDR3 α/β data as input instead of CDR3 β information only. Here, we exclusively employed the CDR3 β single-input model from NetTCR-2.0 for a fair comparison, omitting the utilization of the paired CDR3 α/β dual-input model. ERGO [21] applies two encoding methods, Long Short Term Memory (LSTM) acceptor encoding and Autoencoder-based encoding, for predicting TCR-peptide binding. DLpTCR [22] uses a variety of mixed encoding methods, including one-hot, PCP and PCA encoding, and integrates FCN, LeNet-5 and ResNet-20 to predict the interaction between peptide and TCR.

The codes for the aforementioned models is sourced from their respective GitHub repositories.

The models were retrained and evaluated on our dataset. We trained the models on the IEDB dataset [31], which comprises of 76,348 peptide-TCR pairs, using 5-fold cross-validation. The dataset was randomly divided into five equal parts, and the models were evaluated in each part in a rotating manner (Methods). For independent testing, we used data collected from the VDJdb [32], which contains peptide-TCR pairs with confidence scores ranging from 0 to 3, where higher scores indicate more reliable records. Peptide-TCR pairs present in the training data were excluded from the testing data. Throughout our analysis, the area under the curve (AUC) of the receiver operating characteristic (ROC) was used as the metric to estimate model performance. All experiments were performed with default parameter settings.

Revised manuscript (Page 7, Line 241-244):

ImRex [23] and TITAN [7] are two state-of-the-art (SOTA) models for predicting unseen epitopes and TCRs on independent datasets. ImRex relies on an interaction map, which combines the physicochemical properties of both interactors on the amino-acid level, to predict peptide-TCR recognition under different dataset settings. TITAN exploits convolutions with an interpretable attention mechanism to aggregate local information and integrates the modalities, from which binding probabilities are predicted. To compare HeteroTCR directly, we trained and tested it under their respective training and testing data.

Revised manuscript (Page 18, Line 578-584):

4.7 Machine Learning Platform and Hardware

The pre-trained CNN module was implemented with Keras 2.6.0 (<https://keras.io>) using the Tensorflow backend and Python 3.7.0. The Heterogeneous GNN module and the MLP classifier were implemented with PyTorch 1.9.1 (<https://pytorch.org>) using Python 3.7.11. PyG (PyTorch Geometric) [38], a library built upon PyTorch to easily write and train GNNs, was employed for modeling and processing graph-structured data. Metric evaluation was implemented with TorchMetrics package [40]. All deep learning models were trained on a single NVIDIA A40 graphics card.

C. Many readers of Communications Biology will certainly be unfamiliar with GNNs, and additional motivation and description may be helpful. For instance, in what other areas have GNNs been leveraged for prediction? Have they been used previously in biology or protein interaction studies? Do the authors cite a good background reference for interested readers? Can author provide some intuition to the lay person about how the heterogenous GNN module incorporates a new unknown sequences or peptides in the prediction task? It is somewhat clear how the graph could be formed from training data but how are new unlabeled data incorporated?

Response:

- (1) We appreciate your suggestion to provide additional motivation and description of Graph Neural Networks (GNNs) in the manuscript. We will certainly address these points and include more background information on GNNs, particularly in their applications to biology and protein interaction studies (Page 2, Line 64-68).
- (2) We appreciate your inquiry regarding the integration of new, unknown sequences or peptides in our prediction task. The heterogeneous GNN module is designed to generalize well to unseen data. During training, the model learns the underlying patterns and relationships from labeled

sequences. When faced with new, unlabeled data, the trained model can leverage the learned features and relationships to make predictions. Essentially, the GNN module can extend its knowledge from the training set to make informed predictions for novel sequences or peptides, enabling it to handle previously unseen data with a degree of accuracy.

A. In simple terms, during training, we input two sequences, namely peptide and TCR, and then learn the parameters of the neural network based on given labels, progressively adjusting the output to approximate the labels. In the graph neural network, the entire training set forms a graph, where the connections between any two points indicate the presence or absence of binding. Our focus is on the trained network parameters, which contain the underlying patterns and relationships (between-type interaction and within-type similarity) derived from labeled sequences.

B. When faced with new, unlabeled data, we input two sequences as well, using the trained parameters to predict the binding probability between them. In the graph neural network, the entire test set forms another graph, where the connections between points represent peptide-TCR pairs that the test set wants to predict for binding. The previously trained network parameters are then applied to predict the outcomes.

Considering the consistency throughout the two processes, providing further detailed, layperson-oriented explanations in the text may result in redundancy. However, we will carefully examine our manuscript to ensure that there are no areas where additional explanations are needed, and we will strive to maintain logical clarity in our expressions. Thank you for your understanding.

Revised manuscript (Page 2, Line 64-68):

Graph neural networks (GNNs) have made remarkable strides in recent years, establishing themselves as essential tools for a range of graph-based applications. Notably, they have demonstrated success in predicting chemical stability [26], forecasting protein solubility [27], modeling protein-protein interaction prediction [28], and exploring drug-target interactions [29]. Therefore, motivated by these successes in the use of GNNs, we introduce HeteroTCR (Fig. 1), a Heterogeneous Graph Neural Network based SPM that utilizes only CDR3 β sequence information from TCRs and peptides to improve predictive accuracy for peptide-TCR recognition across various datasets.

D. The study mentions “ablation studies” and “visualization” substantiating the essential role of the GNN in finding feature underlying binding process. In this dimension the manuscript could be strengthened. First, the visualization based on tSNE are hard to interpret and it’s not clear if the claims could be falsified based on their inspection alone.

Response: Thank you for your valuable comments and suggestions. In the revised manuscript, we introduce additional numerical metrics that measure the distances of the colored points in the GNN embedded space compared to those in the CNN embedded space, as illustrated in Fig. 5c. Fig. 5d, previously labeled as Fig. 5c in the initial version of the manuscript, depicts the AUC of models with and without the heterogeneous GNN module. This figure underscores that positive and negative samples are more easily separable in the GNN embedded space than in the CNN embedded space. For detailed information, please refer to our response to Reviewer #2 Comment 2.

Additionally, the authors use “ablation studies” to mean removal of components of the AI system to assess changes in performance. This is rational, but this word choice could also be confusing to biologists who may expect “ablation” to mean changing predicted residues of a peptide or TCR to test binding predictions.

Response: We appreciate your observation regarding the term “ablation studies” and understand the potential for confusion, especially for biologists familiar with a different context. To address the potential confusion, we will make efforts to clarify this terminology in the manuscript (Page 9, Line 276-281). We aim to enhance the clarity of our descriptions to ensure a better understanding for readers from diverse backgrounds.

Revised manuscript (Page 9, Line 276-281):

To emphasize the advantages of HeteroTCR, we conducted ablation studies by establishing a baseline model. In computational terms, “ablation studies” refer to the systematic removal of components from the computational model to assess its impact on overall performance, rather than modifying predicted residues of peptides or TCRs. In our baseline model, we excluded the Heterogeneous GNN module, enabling us to assess the module’s contribution to enhancing both between-type (peptide-TCR) interaction and within-type (TCR-TCR or peptide-peptide) similarity. Models were trained on IEDB data using four types of data splitting methods, with McPAS-TCR as the validation dataset. Further evaluation took place independently on VDJdb with a confidence score greater than or equal to 0.

While this experimental work may be out of scope for a computational effort, could the authors use their tool to generate specific falsifiable predictions about which residues could be changed to abrogate TCR-pMHC reactivity? Alternatively, can the “model extracted representations” be visualized in any way to enable human interpretable understanding CDR3 feature essential for peptide recognition?

Response: Thank you for your valuable comments and suggestions. This may indeed be out of scope for a computational effort. We attempted to systematically remove amino acids at specific positions in all TCR or peptide sequences to identify which residues could be essential for TCR-pMHC reactivity. However, our encoding involves padding with zeros at the end to ensure uniform input lengths for model training. When removing amino acids at specific positions, discrepancies arise. For example, the 15th position in the first TCR may be zero-padded, while the 15th position in the second TCR contains an amino acid embedding. This leads to a situation where, by removing amino acids at specific positions, the model may develop biases towards considering the amino acids at the end of the sequence as less important.

E. The authors assert that their method outperforms existing models. In addition to comparison made, can the authors contextualize the gain in performance?

Response: Thank you for your insightful comments and questions. HeteroTCR captures within-type (TCR-TCR or peptide-peptide) similarity information and between-type (peptide-TCR) interaction insights for predictions. For pre-trained CNN module, Montemurro et al [5] propose that the ability of classification in CNN-based prediction models is driven by the representation in the max-pooled CNN layer. Therefore, we extract the numeric embeddings of TCRs and peptides from the max-pooled CNN layer of the pre-trained CNN module for subsequent module. For

Heterogeneous GNN module, we use a set of aggregator functions to learn aggregate feature information from K^{th} -order neighborhoods of a node. A given peptide/TCR can extract between-type (peptide-TCR) interaction information from the 1st- and 3rd-order TCRs/peptides, and obtain within-type (TCR-TCR or peptide-peptide) similarity information from the 2nd-order peptides/TCRs. Therefore, our model can capture more interaction information, theoretically surpassing other models.

Given that this is a very challenging prediction task with a limited amount of training data, can the authors clarify whether the performance achieved (AUC = .6535, strict data splitting, figure 3) might allow in terms of any practical application?

Response: Thank you for your questions. While it is acknowledged that our model may not uncover all possible interactions, it is important to note that in practical applications, exhaustive identification of all interactions may not be necessary. Our primary objective is to identify TCR-peptide pairs that can elicit an immune response. By prioritizing the prediction of the model, we can significantly reduce the cost associated with subsequent experimental validations. The vast number of potential interacting pairs offers a substantial pool, and by identifying only a few, we can potentially address therapeutic needs. It is not imperative to identify every possible interaction, but rather those with a high likelihood of clinical relevance.

Does the present work provide some indication on the amount or quality of training data that would be necessary for the model to achieve a higher level of accuracy? For instance, what if half as much training data had been used? What sort of data would be most valuable for increasing generalizable prediction?

Response: Thank you for your question. To address this issue, we have conducted experiments to evaluate the impact of varying the number of peptides in the training dataset. Our findings suggest that diverse and representative samples, encompassing a wide range of scenarios, are of utmost importance for enhancing generalizable prediction. For further details, please refer to the following revised manuscript, Fig. 7, and Supplementary Table 10.

Revised manuscript (Page 12, Line 354-370):

Fig. 7. Performance of HeteroTCR across different number of subsampled peptides based on strict-based data

sets. The shaded area depicts the mean \pm the standard deviation over five repeated experiments. In the validation dataset and testing dataset, the number of pairs for different numbers of subsampled peptides is 784 and 1208, respectively. For different numbers of subsampled peptides (ranging from 100 to 559), the corresponding numbers of pairs in the training dataset are 7180, 22440, 40932, 52636, 69816, 76348, respectively.

Additionally, we investigated the impact of the number of peptides on model performance. In this analysis, we trained models on IEDB dataset using strict-based datasets, and utilized McPAS-TCR as the validation dataset for parameter selection. Evaluation was performed on VDJdb with a confidence score greater than or equal to 0. In the experiments, the initial training dataset consisted of 559 unique peptides. We randomly subsampled them to reduce the quantity to 100, 200, 300, 400, and 500. Our observations revealed that as the number of unique peptides increased, the model performance improved. When the number of subsampled peptides reached 500, the model AUC reached 0.6471, and further increases showed diminishing returns (additional details in Supplementary Table 10). Our findings suggest that diverse and representative samples, encompassing a wide range of scenarios, are of utmost importance for enhancing generalizable prediction.

Does the heterodimeric nature of the TCR receptor place expected limits on any method based on the CDR3 alone?

Response: We appreciate the reviewer's insightful comment. Numerous studies have previously delved into this matter, consistently indicating that leveraging both TCR CDR3 chains is more effective than relying solely on the β chain. Moreover, incorporating the complete TCR chain information has been demonstrated to be superior to focusing exclusively on the CDR3 region. Our framework is designed for broad applicability, allowing for the flexible inclusion of α chain, complete TCR sequence or MHC-type information in the network for both training and prediction. This design permits the framework to fully capitalize on additional sequence information, enhancing model performance once such data become available.

F. While the tool may represent a computational advance, the application section 2.6 is very cursory and may not justify the full claims of the tools utility made in the discussion.

Response: We appreciate your feedback. In the revised manuscript, we significantly toned down our claims in Section 2.6. For a more detailed explanation, please see our responses to Specific Comments, particularly Comment 14-16, where we provide a more comprehensive clarification on Section 2.6.

Specific Comments

Comment 1:

The author suggest that their code is made publicly available. The link in the manuscript is a generic link to GitHub and does not provide access for reviewers to evaluate. Code for this work should be made available to reviewers in advance and must be made available prior to publication.

Response: Thank you for pointing this out. We apologize for any inconvenience caused by the generic GitHub link provided. We have now rectified this and made our code publicly accessible for reviewers. The updated link is <https://github.com/yuzilan/HeteroTCR>.

Comment 2:

Line 70 Please provide additional explanation of how the GNN extracts “within-type” information about “TCR-TCR” or “peptide-peptide”. Please provide the reader with intuition on how the training data (typically a TCR to a pMHC ligand assignment) also provides information about “peptide-peptide” similarity? Clarify line 123. What is meant by “This implies that TCRs (peptides) associated with the same peptide (TCR) may similarity to some extent, making it possible to capture within-type (TCR-TCR or peptide-peptide) similarity”. I found this sentence confusing. Does this require training data where two peptides are recognized by the same CDR3b sequence? If so how common is this in the training data?

Response: Thank you for your valuable feedback. We appreciate your insightful comments and suggestions. We will address the concerns raised in your inquiry:

- (1) The content in the Introduction Section regarding “how the GNN extracts within-type information about TCR-TCR or peptide-peptide” is a concise overview of some of HeteroTCR’s contributions. Therefore, it is not feasible to provide an extensive explanation around line 70. Detailed explanations about the model’s specific architecture, as well as discussions on between-type (peptide-TCR) interaction and within-type (TCR-TCR or peptide-peptide) similarity, are presented in sections 2.1, 4.3, 4.4, 4.5, and Fig.1.
- (2) Line 123: We acknowledge the confusion caused by the sentence, and we will revise it to provide a clearer explanation (Page4, Line 133-134). Our intention is to convey that the binding relationship between peptides and TCRs is many-to-many. TCRs (peptides) associated with the same peptide (TCR) may exhibit some degree of similarity. These established conditions consequently enable the capture of within-type (TCR-TCR or peptide-peptide) similarity. Therefore, there is no requirement for “two peptides are recognized by the same CDR3b sequence” in the training data; the training dataset imposes no such restrictions.

Comment 3:

Line 150: Why is 0.5 an appropriate threshold for predicting interaction?

Response: Thank you for raising this question. The choice of a 0.5 represents a balanced decision point, where predictions with a probability greater than or equal to 0.5 are classified as positive, and those below 0.5 are classified as negative. When the prediction range is from 0 to 1, this threshold is often used as a default.

Comment 4:

Line 156: word "ratating" perhaps “rotating”

Response: Thank you for pointing this out. We have corrected it to “rotating” in the revised manuscript.

Comment 5:

Fig 2: consider removing gray line (trend in model performance, seems irrelevant).

Response: Thank you for your thoughtful feedback. We have removed the gray line.

Comment 6:

Fig 2: What are the repeated measures in each boxplot, was 5-fold data partition of the training data

used to create a separate model that was then applied to the VDJdb testing data?

Response: Thank you for your inquiry. In each boxplot, the repeated measures refer to the performance metrics obtained across the 5-fold cross-validation of the training data. The dataset was randomly divided into five equal parts. In each iteration, four parts were used as the training set, and the remaining one part served as the validation set. This process was repeated five times, resulting in the creation of five models. Subsequently, we evaluated the performance of these five models on an independent testing dataset (VDJdb) and obtained five AUC metrics. Details are elucidated in Section 4.1 of our manuscript.

Comment 7:

Line 179: what does “alone” in “TCRs present alone in the training set” mean. Does this just mean that exact TCR (CDR3b) sequences present in the training data were removed from the testing dataset? Please clarify the data-splitting rule description Line 178-182.

Description in methods is clearer.

Response: Thank you for pointing this out. We apologize for the lack of clarity in our descriptions of the data. We have revised the manuscript to provide a clearer description of the data-splitting rule in Lines 200-204.

Revised manuscript (Page 6, Line 200-204):

To further demonstrate the generalizability of HeteroTCR, we evaluated models based on four different types of data splitting methods (Methods), namely pair-based, TCR-based, antigen-based, and strict-based data sets. For the pair-based data sets, the peptide-TCR pairs present in the training dataset were removed from the testing dataset. For TCR-based data sets, peptide-TCR pairs involving TCRs present in the training dataset are excluded from the testing dataset. Similarly, for antigen-based data sets, peptide-TCR pairs involving peptides appearing in the training dataset are excluded from the testing dataset. In the case of strict-based data sets, peptide-TCR pairs involving either TCRs or peptides present in the training dataset are excluded from the testing dataset.

Comment 8:

Fig 3: An interesting feature of Figure 3 is the low variance in AUC for HeteroTCR vs other methods. What might explain this?

Response: Thank you for your insightful observation. The lower variance in AUC for HeteroTCR compared to other methods may be attributed to the robustness of the GNN model, which consistently performs well across diverse datasets and demonstrates a more stable predictive performance. We have incorporated this explanation into the revised manuscript (Page 7, Line 225-228).

Revised manuscript (Page 7, Line 225-228):

Additionally, the lower variance in AUC for HeteroTCR compared to other methods was likely attributed to the robustness of the GNN model, which consistently performs well across diverse datasets and demonstrates a more stable predictive performance. This stability is a noteworthy characteristic that enhances the reliability and consistency of HeteroTCR’s predictive capabilities.

Comment 9:

Line 247: “Further evaluation took place independently on VDJdb with a confidence score greater than or equal to 0” – Since 0 is the lowest confidence score, doesn’t this just imply all of VDJdb was used regardless of confidence score?

Response: Thank you for the question. You are right. The intended meaning was to ensure that we considered data points with a confidence score greater than or equal to 0, encompassing the entire dataset. It is important to note that the dataset underwent the four-step processing outlined in Section 4.1. We employed the confidence score for partitioning, as it is an inherent value in the VDJdb.

Comment 10:

Line 259 -How is model representation embedded with tSNE, could more details be provided.

Response: Thank you for your inquiry. t-SNE (t-distributed Stochastic Neighbor Embedding) is a widely used algorithm for data dimensionality reduction and visualization. Its primary objective is to map high-dimensional data to a lower-dimensional space while preserving the local relationships between data points. t-SNE achieves this by modeling probability distributions to maintain similarity between points in both high and low-dimensional spaces.

What is the dimension of data being embedded? Figure 5 shows with and without GNN module, but it is very hard to understand what exactly the “model-extracted representations of peptide-TCR pairs” are. Consider a supplementary diagrammatic figure that could provide more detail than Fig 1 on the dimensions of vectorized representations with and without GNN?

Response: Thank you for your inquiry. Firstly, the TCR sequences were zero-padded to a maximum length of 20, while the peptide sequences to a maximum length of 15. We then encoded AA sequences of TCRs and peptides using the BLOSUM50 matrix, in which each AA is represented as the score for substituting the AA with all the 20 amino AAs. Hence, the BLOSUM encoding scheme maps a sequence with a length of l to an array with a size of $l \times 20$. Secondly, the TCR sequences and the peptide sequences were separately deconvoluted by different convolution kernels with kernel size $\{1, 3, 5, 7, 9\}$, in which different features were integrated through different convolution kernels to filter the whole sequence. For each kernel size, the convolutional output was max-pooled and the resulting feature vectors were concatenated in a single vector with 160 entries (80 for each input sequence). Thirdly, the 160-dimensional vector was fed into a dense layer of 32 hidden neurons and the output consists of one single neuron, giving the probability of a peptide-TCR pair to bind (Fig. 1 left). Finally, the 160-dimensional max-pooling layers of the pre-trained CNN module were extracted as numeric embeddings of TCRs and peptides to input the Heterogeneous GNN. The input of Heterogeneous GNN is an 80-dimensional numeric embedding of TCR and an 80-dimensional numeric embedding of peptide, and the output is a 1024-dimensional TCR vector and a 1024-dimensional peptide vector. We provide a simplified diagram here, although we do not recommend adding it to the supplementary information, as detailed information has already been provided in Sections 4.3 to 4.5 and Fig. 1.

Comment 11:

It is hard to see “distinct boundary” in Fig 5a (right). Maybe a close up figure of the relevant points or another term is more appropriate.

Response: Thank you for your insightful comments on Figure 5a (right) in our manuscript. We appreciate your feedback regarding the term “distinct boundary”. We have explored alternative terminology that better encapsulates the separation between the data points.

Revised manuscript (Page 10, Line 321-323):

Notably, using numeric representations extracted by the full model, cognate TCRs for a specific antigen clustered more tightly compared to a model without the Heterogeneous GNN module. Additionally, with the Heterogeneous GNN module, we observed a better separation between the orange dots (representing positive samples, indicating peptide-TCR interactions) and the purple dots (indicating negative samples, denoting no peptide-TCR interactions).

Comment 12:

Line 290: the word “decrease” is likely misspelled

Response: Thank you for pointing this out. We have corrected it in the revised manuscript.

Comment 13:

Line 286, why use Levenshtein ratio instead of distance? Figure 5d is confusing, one would expect more data at further Levenshtein distance but the red line suggests that there is less data at further distance? Is this correct? Also, it would be much more interpretable to a biologist to explain that the model can generalize even to TCRs X-X “edit distance” away from the nearest training set TCR? The ratio is explained in methods, but can more information be provided for interpretation.

Response: We appreciate your insightful comments. The choice of using Levenshtein ratio instead of distance was based on its ability to account for variations in the lengths of sequences, providing a normalized measure.

(1) Levenshtein distance, also known as edit distance, describes the minimum number of operations required to transform one string into another. These operations include insertion, deletion, and substitution. For example:

```
>>> import Levenshtein
>>> Levenshtein.distance('abc', 'ac')
1
>>> Levenshtein.distance('kitten', 'sitting')
3
```

kitten → ① → sitten → ② → sittin → ③ → sitting

① ‘k’ to ‘s’ ② ‘e’ to ‘i’ ③ add ‘g’

(2) Levenshtein ratio is explained in Section 4.6. In simple terms, Levenshtein ratio = (sum - ldist) / sum, where sum represents the total length of the two strings, and ldist is the “real minimal edit distance”, distinct from the edit distance (Levenshtein distance). In the edit distance, each operation is counted as 1, while in ldist, insertion and deletion operations are still counted as 1, but substitution is counted as 2.

```
>>> import Levenshtein
>>> Levenshtein.ratio('abc', 'abc')
1.0
>>> Levenshtein.ratio('abc', 'ab') # (5-1)/5 = 0.8
0.8
>>> Levenshtein.ratio('abc', 'abd') # (6-2)/6 = 2/3 = 0.666
0.6666666666666666
```

In summary, Levenshtein distance measures the dissimilarity between two strings, with lower scores indicating greater similarity. On the other hand, the Levenshtein ratio calculates the similarity between two strings, where higher scores indicate greater likeness. Therefore, the two metrics are inversely proportional to some extent. In Fig. 5d, the x-axis ranging from 0.30 to 0.24 in Levenshtein ratio represents decreasing similarity. As the data in the dataset becomes less alike (increasing distance), similar data is gradually filtered out, resulting in fewer instances at greater distances. Consequently, if one interprets Levenshtein ratio according to Levenshtein distance, the results would indeed be opposite.

We believe that the information and algorithms provided are sufficiently detailed. The logic of the proportional and inverse relationship needs to be understood independently, and hence, we do not recommend adding more information to the manuscript. Thank you for your understanding.

Comment 14:

Consider an alternative term for “Real-life validation”, perhaps something like “GNN binding predictions correlate with experimentally derived TCR pMHC affinity”

Response: Thank you for the great suggestion. We changed the title for the subsection to “HeteroTCR binding predictions correlate with experimentally derived pMHC-TCR affinity” to better convey the intended meaning.

Comment 15:

Overall, section 2.6 is extremely brief and somewhat confusing

Response: We apologize for the lack of clarity in this section. In the revised manuscript, we have provided additional details to enhance clarity.

Revised manuscript (Page 12, Line 374-379):

We download single-cell datasets from the 10x Genomics Chromium Single Cell Immune Profiling platform [37], consisting of samples from four donors. The binding specificity of each CD8+ T cell was assessed using 44 different pMHC. We validated the predicted binding probability between TCRs and peptides by assessing the expected impact of TCRs with higher pMHC affinities. We quantified the pMHC-TCR affinities by calculating the UMI count for the specific pMHC in each T cell. In cases where a cell exhibited multiple specificities, we considered it specific only for the pMHC with the highest UMI count. We calculated the Spearman’s rank correlation coefficient and find a positive correlation between the binding affinities and the predicted binding probabilities for the peptide-TCR pairs (Fig. 8). In other words, the higher the strength of interactions between TCRs and peptides are, the higher the predicted binding probabilities are.

Comment 16:

Correlation between GNN predicted score and binding affinity (which is probably noisily measured in this context) is low but positive. Is the spearman correlation based on the bins or all of the data. Why not show the data as a scatter plot? What do the error bars in Fig 6 indicate?

Response: Thank you for your questions. The Spearman correlation is based on all of the data, encompassing the entire dataset rather than specific bins.

(1) Scatter Plot vs. Bar Plot: Initially, we experimented with scatter plots to represent the data. However, due to the complexity and noise in the measurements, the scatter plots were visually cluttered and challenging to interpret (Refer to the figure below). Considering both aesthetics and clarity, we found that a bar plot, implemented using the seaborn==0.11.2 library in Python, was the most visually appealing and effective way to convey the data.

(2) The error bars in Fig. 6 indicate the 95% confidence interval for the estimation of the mean of each bins. The upper and lower limits of the error bars correspond to the boundaries of the 95% confidence interval for the respective data.

Comment 17:

Line 331: The authors should be careful not to overstate the utility of existing tool and extent of validation. No effort was made to identify neo-epitopes based on HeteroTCR predictions.

Response: Thank you for your insightful feedback. To tone down the statements, we revised the manuscript as follows.

Revised manuscript (Page 14, Line 411-415):

Moreover, HeteroTCR exhibits the capability to predict candidate peptides that may potentially bind to antigen-specific TCR sequences. We verified HeteroTCR's interaction prediction accuracy using immunogenic antigen datasets from the 10x the Genomics Chromium Single Cell Immune Profiling platform. While the model is promising in aiding the identification of immunogenic neo-epitopes, experimental efforts were required to further validate the predictions of HeteroTCR.

Reviewers' comments:

Reviewer #1 (Remarks to the Author):

The authors successfully addressed all the comments in the revision and the manuscript quality improved.

Reviewer #3 (Remarks to the Author):

NOTE: Please see my full detailed review of the revised manuscript and rebuttal letter attached and as a well-formatted word document.

Reviewer Response to Rebuttal Letter and Revised Manuscript

Yu et al. Manuscript: HeteroTCR: A heterogeneous graph neural network-based method for predicting peptide-TCR interaction

The authors addressed my initial concern about the lack data and code availability and information about models and ML tools used (Line 578-584). The authors now include a GitHub link to their code and data, which is well organized and will allow other interested researchers to attempt to reproduce and build on the approaches described.

The authors responded thoroughly to most my prior comments. I appreciate the thoughtful responses, sensitivity analyses, and revisions. I believe the methods in this study are highly innovative and may stimulate new insights and methodologies for TCR data exploration and will be of interest to the field.

After a subsequent review, my only remaining comments focus on section 2.6 and Figure 8. Furthermore, I have a few constructive suggestions about wording in the discussion, that I hope the authors may find useful in their final revisions if the manuscript is accepted for publication.

In section 2.6 Authors conclude: "In other words, the higher the strength of interactions between TCRs and peptides are, the higher the predicted binding probabilities are." Figure 8 needs additional details. I generally agree that the data shows this, but I think some critical details in the data presentation are missing. With some critical methodological details provided in the methods and additions to Figure 8 caption, I think the section would be greatly improved.

Comment 1: Methods for section 2.6

(Page 12, Line 374-379) of the revised manuscript greatly improve communication of results in section 2.5, however, there is no main methods section corresponding to the results section 2.6. to match they otherwise rigorous methodological description provided for other results sections.

In such a methods section:

1.1 Authors could provide detail where raw data was actually accessed. E.g., were data accessed here: <https://zenodo.org/records/6952657>

1.2 How were binding strength estimates for multiple droplets with identical TCRs sharing the same sequence estimated? Median, mean, other?

1.3 Taking "cases where a cell exhibited multiple specificities, we considered it specific only for the pMHC with the highest UMI count". Does this mean that data were dropped for weaker interactions in Figure 8 analysis. If so, please state so clearly? Consider why would these secondary interactions not also be useful for testing relationship between binding strength prediction and observation? They would be expected to have weak prediction and weak binding strength. Specificity was not considered

in this results section?

1.4 Does the manuscript text clearly describe how bin boundaries were defined? From GitHub it seems discrete cut-offs at 0.1 probability increments. If so, state this logical choice in the methods.

https://github.com/yuzilan/HeteroTCR/blob/d684a6b4b8785fbb2a39d7e38753ae8b5f1e616e/data/original_10x/get_x_y.py#L34C1-L55C28

1.5 Where there any thresholding or important data filtering steps required, if so state in methods (e.g., https://github.com/yuzilan/HeteroTCR/blob/main/data/original_10x/filter_cell.py)

1.6 Did the authors consider a sensitivity test looking only at expanded clones where "binding strength" might be more robustly estimated across droplets with identical receptor sequences compared to more noisy singletons?

1.7 Since affinity has a particular biochemical meaning which is only approximated by droplet-seq assay with DNA barcoded pMHC dextramer data used in section 2.6, The authors may wish to consider if it would be more appropriate to replace "affinity" in the y-axis of Figure 8 with "log10(UMI)" or "estimated pMHC-TCR binding strength (log10(UMI))".

1.8 Authors could include in the supplement to this paper a version of the scatter plot provided in their response to Comment 16. However, it would be more useful if the y-axis is log transformed to match the main figure. Experimentalists, seeking to use this tool a guidepost for selection, will certainly appreciate seeing the range and distribution of observed versus predicted binding strengths shown well in the scatter plot but not clearly in bar plots

Comment 2: Figure 8

In their rebuttal authors state:

"We calculated the Spearman's rank correlation coefficient and find a positive correlation between the binding affinities and the predicted binding probabilities for the peptide-TCR pairs (Fig. 8)."

Were Spearman rank correlations computed on the full dataset regardless of binning? Or was the spearman rank correlation test performed on bin summary measures (i.e., median bin score) ?Please clarify in methods or figure caption .

2.1 In caption, specify what the error bars on each bar represent in Figure 8.

2.2 In caption. For group comparisons (i.e., bin 9 vs bin 10), if comparison between log10 UMI values between bind 9 and 10 is made was a Wilcoxon test performed? If so, state clearly in figure caption.

2.2 In general, boxplots would be far more informative than barplots, see constructive suggestions below for improving the quality of data visualization or providing a useful supporting figure for data underlying Figure 8.

Figure 8:

Constructive suggestion for alternative data presentation:

...

```
require(ggplot2)
require(dplyr)
a=readr::read_csv('./HeteroTCR/data/original_10x/donor1/affinity.csv')
a$peptide = extracted_elements <- sapply(strsplit(a$peptide, "_"), function(x) x[2])
# Not clear how results are aggregated over clones with identical cdr3a,cdr3b
a = a %>% group_by(cdr3a, cdr3b, peptide) %>%
summarise(affinity = median(affinity))
p=readr::read_tsv('./HeteroTCR/data/original_10x/donor1/pred.tsv')
aff_pred = dplyr::left_join(a, p, by = c("cdr3b"= "cdr3", "peptide"="peptide"))
aff_pred$bin = cut(aff_pred$probability, breaks = seq(0,1,1))
```

```

p1 = ggplot(aff_pred,
aes(y = affinity,
x = probability))+
geom_point(size = .1) +
ggtitle("Donor1") +
ylab("log10(UMI)") +
xlab("predicted binding probability") +
facet_grid(~bin, scale = "free_x")
p1_box = p1$data %>%
ggplot(aes(x = bin, y = affinity)) +
geom_boxplot(outlier.size = .1) + scale_y_log10()

gridExtra::grid.arrange(p1+scale_y_log10() + theme(axis.text= element_text(angle = 90)),
p1_box)
` ``

```

Comment 3

I appreciate that the authors added a caveat in the discussion, but I think the authors should consider revising the second to last discussion paragraph.

The authors currently write:

“Moreover, HeteroTCR exhibits the capability to predict candidate peptides that may potentially bind to antigen-specific TCR sequences. We verified HeteroTCR’s interaction prediction accuracy using immunogenic antigen datasets from the 10x Genomics Chromium Single Cell Immune Profiling platform. 414 While the model is promising in aiding the identification of immunogenic neo-epitopes, experimental efforts 415 were required to further validate the predictions of HeteroTCR. ”

I believe the authors may wish to state something along the lines of the following --

“We tested for correlation between HeteroTCR derived pMHC-TCR interaction scores and binding strength predictions inferred in publicly available 10x Genomics data from single cell droplet sequences in the presence of DNA barcoded pMHC dextramers. While the model is promising in aiding the identification of immunogenic epitopes, a wide range of binding strengths were observed even among TCR with the highest HeteroTCR GNN-inferred interaction scores. This underscores that HeteroTCR may have utility to prioritize candidate TCRs of clinical relevance, but binding prediction will require experimental validation.”

Comment 4

Line 369-370 “Our findings suggest that diverse and representative samples, encompassing a wide range of scenarios, are of utmost importance for enhancing generalizable prediction”.

The authors might consider clarifying what is meant by this sentence. For instances what does diverse samples “encompassing a wide range of scenarios” refer to in the context of TCR biology? For instance, do the authors mean that it is important to have training data including many peptides-TCR interactions identified in HLA-diverse human populations with varied immunological exposures?

Comment 5

The authors infer from the new Figure 7, that “further increases in peptides showed diminishing returns (additional details in Supplementary Table 10). The sensitivity analysis suggests that the

model performs better when trained with more data from more peptides, however, it is not clear to me that a plateau in performance has been reached, thus the authors may wish to revise statement about "diminishing returns" and rather emphasize the clearer finding that HeteroGNN improved with greater input data.

--end--

Point-by-point Response to Reviewer's Comments

We sincerely appreciate the diligent review conducted by the esteemed reviewers and express our gratitude for their insightful comments and constructive suggestions. In the sections below, we present a detailed point-by-point response that includes summaries of changes made in response to these insightful comments. The original reviewers' comments are colored in black, and our responses are provided in blue. The changes in the manuscript are highlighted in yellow.

Reviewer #3:

The authors addressed my initial concern about the lack data and code availability and information about models and ML tools used (Line 578-584). The authors now include a GitHub link to their code and data, which is well organized and will allow other interested researchers to attempt to reproduce and build on the approaches described.

The authors responded thoroughly to most my prior comments. I appreciate the thoughtful responses, sensitivity analyses, and revisions. I believe the methods in this study are highly innovative and may stimulate new insights and methodologies for TCR data exploration and will be of interest to the field.

After a subsequent review, my only remaining comments focus on section 2.6 and Figure 8. Furthermore, I have a few constructive suggestions about wording in the discussion, that I hope the authors may find useful in their final revisions if the manuscript is accepted for publication.

In section 2.6 Authors conclude: "In other words, the higher the strength of interactions between TCRs and peptides are, the higher the predicted binding probabilities are." Figure 8 needs additional details. I generally agree that the data shows this, but I think some critical details in the data presentation are missing. With some critical methodological details provided in the methods and additions to Figure 8 caption, I think the section would be greatly improved.

Response: Thank you very much for your kind words and positive feedback on the manuscript. Your thorough examination has significantly contributed to enhancing the quality and clarity of our manuscript. In the subsequent responses to each comment, we have rewritten Section 2.6 as per your request and included the methodological description (Section 4.7).

Comment 1: Methods for section 2.6

(Page 12, Line 374-379) of the revised manuscript greatly improve communication of results in section 2.5, however, there is no main methods section corresponding to the results section 2.6. to match they otherwise rigorous methodological description provided for other results sections.

Response: We appreciate your feedback. In the revised manuscript, we have taken great care to comprehensively rewrite Section 2.6, ensuring that it now includes a thorough and rigorous methodological description (Section 4.7).

Revised manuscript (Page 12, Line 372-395):

2.6 HeteroTCR binding predictions correlate with experimentally derived pMHC-T cell binding fractions

To assess the performance of the model from an alternative angle, we investigated whether the predictions of the HeteroTCR model correlate with experimentally derived pMHC-T cell binding fractions (Methods). The data utilized in our study was generated from the 10x Genomics Chromium

Single Cell Immune Profiling platform [37] (Methods). We analyzed single-cell datasets containing profiles of CD8+ T cells specific to 44 different pMHC multimers, sourced from four healthy donors. The binding specificity between each T-cell and tested pMHC was quantified by counting the number of unique molecular identifier (UMI) sequences associated with each specific pMHC in the T cell. After data curation and the computation of binding fractions (Methods), we proceeded to calculate the Spearman correlation coefficient between HeteroTCR-predicted binding probabilities and binding fractions, i.e. the fraction of a T cell clone bound to a specific TCR.

Figs. 8a-d illustrate the correlation between the predicted binding probabilities of HeteroTCR and binding fractions. Notably, the dataset includes cases where the binding fraction is 0, illustrating an absence of binding specificity. A positive correlation observed in the figures indicates the association between the model's predicted probabilities and the binding events. In Figs. 8e-h, the correlation between the two variables in data excluding instances where the binding fraction equals 0. Consequently, the figures exclusively comprise data with binding specificity, and a positive correlation depicts that the predicted binding probabilities of HeteroTCR are associated with binding strength in instances where binding occurs.

Fig. 8. Predicted binding probability is positively correlated with binding fractions. a-d, Binding predictions for donors 1-4 correlate with binding fractions, where binding fractions include cases equal to 0. The number of data for each donor is 29495, 62956, 41496, and 13430, respectively. e-h, Binding predictions for donors 1-4 correlate with binding fractions, where binding fractions exclude cases equal to 0. The number of data for each donor is 9909, 24156, 13114, and 4792, respectively.

Revised manuscript (Page 18, Line 586-611):

4.7 Data curation of the 10x Genomics platform and calculation of binding fractions

The data utilized in our study was generated from the 10x Genomics Chromium Single Cell Immune Profiling platform (<https://www.10xgenomics.com/resources/datasets>). The raw data is accessible for download at <https://zenodo.org/records/6952657>. Closely examining four single-cell datasets, we analyzed profiles of CD8+ T cells specific to a highly multiplexed panel consisting of 44 different pMHC multimers, alongside 6 negative control pMHC multimers, sourced from four healthy donors. The binding specificity between T cells and each pMHC complex was quantified using the UMI counts as a binding indicator.

In general, a T cell typically expresses only one pair of functional TCRs, and thus, we selectively retain clones of T cells expressing a single pair of TCR α and TCR β chains. We focused

only on expanded clones, as the UMI counts of each cell were inherently noisy due to dropouts and high variances in single-cell experiments. Consequently, we opted to utilize the binding fraction of each T cell clone with the same TCR as a measure of antigen affinity to a TCR. The binding fraction of a clone is determined by the following formula:

$$\text{Binding Fraction} = \frac{m}{n} \quad (6)$$

where m represents the number of T cells within the clone exhibiting a higher UMI count for a given antigen compared to the maximum UMI count among the 6 negative control antigens, and n denotes the clone size, i.e., the number of T cells with identical TCRs. It is noteworthy that instances where m equals 0 are only considered in the following two scenarios: when the UMI count of T cells within the clone for a given antigen is less than the maximum UMI count among the 6 negative control antigens, or when the UMI count of T cells within the clone for a given antigen equals the maximum UMI count among the 6 negative control antigens, and both are non-zero. Specifically, we do not consider cases where m equals 0 when the UMI count of T cells within the clone for a given antigen is 0, and the maximum UMI count among the 6 negative control antigens is also 0. This omission is due to the sparse nature of the original data, characterized by numerous zero values, leading to excessive noise and inclusion of many meaningless data points. Finally, we calculated the Spearman correlation coefficient between predicted binding probabilities and binding fractions.

In such a methods section:

1.1 Authors could provide detail where raw data was actually accessed. E.g., were data accessed here: <https://zenodo.org/records/6952657>

Response: Thank you for your valuable suggestion. The raw data used in our study was indeed accessed from the specified source: <https://zenodo.org/records/6952657>. We have updated the manuscript to explicitly mention this access point in the Section 4.7.

1.2 How were binding strength estimates for multiple droplets with identical TCRs sharing the same sequence estimated? Median, mean, other?

Response: Thank you for your inquiry. We have rewritten Section 2.6 and Section 4.7, ensuring to enhance the transparency of our methodology. In the revised manuscript, we addressed the issue by utilizing the binding fraction instead of the original UMI as a measure.

1.3 Taking “cases where a cell exhibited multiple specificities, we considered it specific only for the pMHC with the highest UMI count”. Does this mean that data were dropped for weaker interactions in Figure 8 analysis. If so, please state so clearly? Consider why would these secondary interactions not also be useful for testing relationship between binding strength prediction and observation? They would be expected to have weak prediction and weak binding strength. Specificity was not considered in this results section?

Response: Thank you for your insightful observation. In our original manuscript, the data were dropped for weaker interactions, which was chosen to prioritize the most prominent interactions and avoid potential noise introduced by weaker interactions. However, in our revised manuscript (Section 2.6 and Section 4.7), we utilized the binding fraction instead of the original UMI as a measure to provide a more comprehensive assessment of the interactions. By considering the binding fraction, we aimed to capture a broader spectrum of interactions, including weaker ones,

which were previously excluded. This adjustment allows us to consider the contribution of all identified specificities, not solely focusing on TCRs specific only for the pMHC with the highest UMI count.

1.4 Does the manuscript text clearly describe how bin boundaries were defined? From GitHub it seems discrete cut-offs at 0.1 probability increments. If so, state this logical choice in the methods. https://github.com/yuzilan/HeteroTCR/blob/d684a6b4b8785fbb2a39d7e38753ae8b5f1e616e/data/original_10x/get_x_y.py#L34C1-L55C28

Response: Thank you for your comment. We have carefully revised Figure 8 according to your request (refer to Comment 2). In the new figure, we have removed the bin boundaries and cut-offs.

Revised manuscript (Page 13, Line 391-395):

Fig. 8. Predicted binding probability is positively correlated with binding fractions. a-d, Binding predictions for donors 1-4 correlate with binding fractions, where binding fractions include cases equal to 0. The number of data for each donor is 29495, 62956, 41496, and 13430, respectively. e-h, Binding predictions for donors 1-4 correlate with binding fractions, where binding fractions exclude cases equal to 0. The number of data for each donor is 9909, 24156, 13114, and 4792, respectively.

1.5 Where there any thresholding or important data filtering steps required, if so state in methods (e.g., https://github.com/yuzilan/HeteroTCR/blob/main/data/original_10x/filter_cell.py)

Response: Thank you for your comment. We have provided a detailed description of the method (Section 4.7) in the revised manuscript. Additionally, the code used for data processing has been uploaded to the provided GitHub repository (https://github.com/yuzilan/HeteroTCR/data/original_10x/data_process.py and https://github.com/yuzilan/HeteroTCR/data/original_10x_bf0/data_process.py).

1.6 Did the authors consider a sensitivity test looking only at expanded clones where “binding strength” might be more robustly estimated across droplets with identical receptor sequences compared to more noisy singletons?

Response: Thank you for your valuable suggestion. We agree with you that the UMI counts of each cell were inherently noisy due to dropouts and high variances in single-cell experiments. Another pertinent factor to consider is the uneven distribution of clone sizes among input T cells in the

original screening experiments (https://pages.10xgenomics.com/rs/446-PBO-704/images/10x_AN047_IP_A_New_Way_of_Exploring_Immunity_Digital.pdf). Given these considerations, relying solely on UMI counts may not provide an accurate measure of binding strength. This is particularly true because larger input clone sizes of a TCR increase the likelihood of relatively high UMI counts, among TCRs with similar read binding strength to an antigen. Consequently, we opted to utilize the binding fraction of each T cell clone with the same TCR as a measure of antigen affinity to a TCR. The binding fraction of a clone is determined by m/n , where 'm' represents the number of T cells within the clone exhibiting a higher UMI count for a given antigen compared to control antigens, and 'n' denotes the clone size, i.e., the number of T cells with identical TCRs.

Revised manuscript (Page 18, Line 586-611):

4.7 Data curation of the 10x Genomics platform and calculation of binding fractions

The data utilized in our study was generated from the 10x Genomics Chromium Single Cell Immune Profiling platform (<https://www.10xgenomics.com/resources/datasets>). The raw data is accessible for download at <https://zenodo.org/records/6952657>. Closely examining four single-cell datasets, we analyzed profiles of CD8+ T cells specific to a highly multiplexed panel consisting of 44 different pMHC multimers, alongside 6 negative control pMHC multimers, sourced from four healthy donors. The binding specificity between T cells and each pMHC complex was quantified using the UMI counts as a binding indicator.

In general, a T cell typically expresses only one pair of functional TCRs, and thus, we selectively retain clones of T cells expressing a single pair of TCR α and TCR β chains. We focused only on expanded clones, as the UMI counts of each cell were inherently noisy due to dropouts and high variances in single-cell experiments. Consequently, we opted to utilize the binding fraction of each T cell clone with the same TCR as a measure of antigen affinity to a TCR. The binding fraction of a clone is determined by the following formula:

$$\text{Binding Fraction} = \frac{m}{n} \quad (6)$$

where m represents the number of T cells within the clone exhibiting a higher UMI count for a given antigen compared to the maximum UMI count among the 6 negative control antigens, and n denotes the clone size, i.e., the number of T cells with identical TCRs. It is noteworthy that instances where m equals 0 are only considered in the following two scenarios: when the UMI count of T cells within the clone for a given antigen is less than the maximum UMI count among the 6 negative control antigens, or when the UMI count of T cells within the clone for a given antigen equals the maximum UMI count among the 6 negative control antigens, and both are non-zero. Specifically, we do not consider cases where m equals 0 when the UMI count of T cells within the clone for a given antigen is 0, and the maximum UMI count among the 6 negative control antigens is also 0. This omission is due to the sparse nature of the original data, characterized by numerous zero values, leading to excessive noise and inclusion of many meaningless data points. Finally, we calculated the Spearman correlation coefficient between predicted binding probabilities and binding fractions.

1.7 Since affinity has a particular biochemical meaning which is only approximated by dropletseq assay with DNA barcoded pMHC dextramer data used in section 2.6, The authors may wish to consider if it would be more appropriate to replace “affinity” in the y-axis of Figure 8 with

“log₁₀(UMI)” or “estimated pMHC-TCR binding strength (log₁₀(UMI))”.

Response: Thank you for your thoughtful suggestion. In the revised manuscript, we have made corresponding modifications in new figures.

1.8 Authors could include in the supplement to this paper a version of the scatter plot provided in their response to Comment 16. However, it would be more useful if the y-axis is log transformed to match the main figure. Experimentalists, seeking to use this tool a guidepost for selection, will certainly appreciate seeing the range and distribution of observed versus predicted binding strengths shown well in the scatter plot but not clearly in bar plots.

Response: Thank you for your comment. We have revised Figure 8 to a scatter plot as per your suggestion (refer to Comment 1.4).

Comment 2: Figure 8

In their rebuttal authors state:

“We calculated the Spearman’s rank correlation coefficient and find a positive correlation between the binding affinities and the predicted binding probabilities for the peptide-TCR pairs (Fig. 8).”

Were Spearman rank correlations computed on the full dataset regardless of binning? Or was the Spearman rank correlation test performed on bin summary measures (i.e., median bin score)? Please clarify in methods or figure caption.

Response: Thank you for your comment. The Spearman correlation coefficients were computed on the full dataset. We have revised Figure 8 based on your following suggestions. In the new figure, we have removed the bins (refer to Comment 1.4).

2.1 In caption, specify what the error bars on each bar represent in Figure 8.

Response: Thank you for your feedback regarding Figure 8. In the revised version, error bars have been removed.

2.2 In caption. For group comparisons (i.e., bin 9 vs bin 10), if comparison between log₁₀ UMI values between bin 9 and 10 is made was a Wilcoxon test performed? If so, state clearly in figure caption.

Response: Thank you for your comment regarding Figure 8. We have revised the figure as per your suggestion. In the updated version, the Wilcoxon test has not been included.

2.3 In general, boxplots would be far more informative than barplots, see constructive suggestions below for improving the quality of data visualization or providing a useful supporting figure for data underlying Figure 8.

Constructive suggestion for alternative data presentation:

```

require(ggplot2)
require(dplyr)
a=readr::read_csv('./HeteroTCR/data/original_10x/donor1/affinity.csv')
a$peptide = extracted_elements <- sapply(strsplit(a$peptide, "_"), function(x) x[2])
# Not clear how results are aggregated over clones with identical cdr3a,cdr3b
a = a %>% group_by(cdr3a, cdr3b, peptide) %>%
  summarise(affinity = median(affinity))
p=readr::read_tsv('./HeteroTCR/data/original_10x/donor1/pred.tsv')
aff_pred = dplyr::left_join(a, p, by = c("cdr3b"= "cdr3", "peptide"="peptide"))
aff_pred$bin = cut(aff_pred$probability, breaks = seq(0,1,1..1))
p1 = ggplot(aff_pred,
  aes(y = affinity,
      x = probability))+
  geom_point(size = .1) +
  ggtitle("Donor1") +
  ylab("log10(UMI)") +
  xlab("predicted binding probability") +
  facet_grid(~bin, scale = "free_x")
p1_box = p1$data %>%
  ggplot(aes(x = bin, y = affinity)) +
  geom_boxplot(outlier.size = .1) + scale_y_log10()

gridExtra::grid.arrange(p1+scale_y_log10() + theme(axis.text= element_text(angle = 90)),
  p1_box)

```

Response: Thank you for your thoughtful suggestions. We truly appreciate your diligence and for providing us with the reference code for data visualization. We have taken your advice and have redrawn Figure 8 accordingly (refer to Comment 1.4). For detailed code reference, please consult our GitHub repository (https://github.com/yuzilan/HeteroTCR/data/original_10x/scatter.R and https://github.com/yuzilan/HeteroTCR/data/original_10x_bf0/scatter.R).

Comment 3

I appreciate that the authors added a caveat in the discussion, but I think the authors should consider revising the second to last discussion paragraph.

The authors currently write:

“Moreover, HeteroTCR exhibits the capability to predict candidate peptides that may potentially bind to antigen-specific TCR sequences. We verified HeteroTCR’s interaction prediction accuracy using immunogenic antigen datasets from the 10x to the Genomics Chromium Single Cell Immune Profiling platform. While the model is promising in aiding the identification of immunogenic neo-epitopes, experimental efforts were required to further validate the predictions of HeteroTCR.”

I believe the authors may wish to state something along the lines of the following --

“We tested for correlation between HeteroTCR derived pMHC-TCR interaction scores and binding strength predictions inferred in publicly available 10x Genomics data from single cell droplet

sequences in the presence of DNA barcoded pMHC dextramers. While the model is promising in aiding the identification of immunogenic epitopes, a wide range of binding strengths were observed even among TCR with the highest HeteroTCR GNN-inferred interaction scores. This underscores that HeteroTCR may have utility to prioritize candidate TCRs of clinical relevance, but binding prediction will require experimental validation.”

Response: Thank you for your thoughtful feedback. We have revised the paragraph to better reflect the points you have raised and provide a clearer explanation of our findings and implications.

Revised manuscript (Page 14, Line 416-422):

Moreover, we investigated the correlation between HeteroTCR’s binding predictions and experimentally derived pMHC-T cell binding fractions using publicly available 10x Genomics data from single cell droplet sequences in the presence of DNA barcoded pMHC dextramers. While the model is promising in aiding the identification of immunogenic epitopes, a wide range of binding strengths were observed even among TCR with the highest HeteroTCR GNN-inferred interaction scores. This underscores that HeteroTCR may have utility to prioritize candidate TCRs of clinical relevance, but binding prediction will require experimental validation.

Comment 4

Line 369-370 “Our findings suggest that diverse and representative samples, encompassing a wide range of scenarios, are of utmost importance for enhancing generalizable prediction”.

The authors might consider clarifying what is meant by this sentence. For instances what does diverse samples “encompassing a wide range of scenarios” refer to in the context of TCR biology? For instance, do the authors mean that it is important to have training data including many peptides-TCR interactions identified in HLA-diverse human populations with varied immunological exposures?

Response: Thank you for your thoughtful review. We have revised the manuscript as follows.

Revised manuscript (Page 12, Line 368-371):

Additionally, we investigated the impact of the number of peptides on model performance. In this analysis, we trained models on IEDB dataset using strict-based datasets, and utilized McPAS-TCR as the validation dataset for parameter selection. Evaluation was performed on VDJDdb with a confidence score greater than or equal to 0. In the experiments, the initial training dataset consisted of 559 unique peptides. We randomly subsampled them to reduce the quantity to 100, 200, 300, 400, and 500. Our observations revealed that the model performance continued to improve with increased input data, suggesting that HeteroTCR benefited from the inclusion of more peptides (additional details in Supplementary Table 10). In summary, our experiments demonstrate that increasing the number of peptides in the training dataset consistently improves the performance of the model, underscoring the importance of diverse antigenic exposures in enhancing predictive accuracy.

Comment 5

The authors infer from the new Figure 7, that “further increases in peptides showed diminishing returns (additional details in Supplementary Table 10). The sensitivity analysis suggests that the model performs better when trained with more data from more peptides, however, it is not clear to me that a plateau in performance has been reached, thus the authors may wish to revise statement

about “diminishing returns” and rather emphasize the clearer finding that HeteroGNN improved with greater input data.

Response: Thank you very much for your constructive suggestions. We have revised the manuscript as follows.

Revised manuscript (Page 12, Line 366-368):

Additionally, we investigated the impact of the number of peptides on model performance. In this analysis, we trained models on IEDB dataset using strict-based datasets, and utilized McPAS-TCR as the validation dataset for parameter selection. Evaluation was performed on VDJdb with a confidence score greater than or equal to 0. In the experiments, the initial training dataset consisted of 559 unique peptides. We randomly subsampled them to reduce the quantity to 100, 200, 300, 400, and 500. Our observations revealed that the model performance continued to improve with increased input data, suggesting that HeteroTCR benefited from the inclusion of more peptides (additional details in Supplementary Table 10). In summary, our experiments demonstrate that increasing the number of peptides in the training dataset consistently improves the performance of the model, underscoring the importance of diverse antigenic exposures in enhancing predictive accuracy.

REVIEWERS' COMMENTS:

Reviewer #3 (Remarks to the Author):

The authors comprehensively addressed my comments.